# Diagnostic Potential of CTRP5 and Chemerin for Coronary Artery Disease: A Study by Coronary Computed Tomography Angiography

**DOI:** 10.3390/diagnostics15020206

**Published:** 2025-01-17

**Authors:** Taha Okan, Cihan Altın, Caner Topaloglu, Mehmet Doruk, Mehmet Birhan Yılmaz

**Affiliations:** 1Cardiology Department, Kardiya Medical Center, 35550 Izmir, Turkey; 2Cardiology Department, Faculty of Medicine, Izmir Economy University, 35550 Izmir, Turkey; drcihanaltin@hotmail.com (C.A.); topalolu@gmail.com (C.T.); 3Izmir Endocrinology Clinic, 35500 Izmir, Turkey; drmehmetdoruk@gmail.com; 4Cardiology Department, Faculty of Medicine, Dokuz Eylul University, 35340 Izmir, Turkey; prof.dr.mbyilmaz@gmail.com

**Keywords:** CTRP5, chemerin, coronary artery disease, coronary computed tomography angiography, Agatston score, coronary artery calcification, atherosclerosis, biomarkers

## Abstract

**Background/Objectives**: As an endocrine organ, adipose tissue produces adipokines that influence coronary artery disease (CAD). The objective of this study was to assess the potential value of CTRP5 and chemerin in differentiating coronary computed tomography angiography (CCTA)-confirmed coronary artery disease (CAD) versus non-CAD. Secondarily, within the CCTA-confirmed CAD group, the aim was to investigate the relationship between the severity and extent of CAD, as determined by coronary artery calcium score (CACS), and the levels of CTRP5 and chemerin. **Methods**: Consecutive individuals with chest pain underwent CCTA to evaluate coronary artery anatomy and were divided into two groups. The CCTA-confirmed CAD group included patients with any atherosclerotic plaque (soft, mixed, or calcified) regardless of calcification, while the non-CAD group consisted of individuals without plaques on CCTA, with zero CACS, and without ischemia on stress ECG. Secondarily, in the CCTA-confirmed CAD group, the severity and extent of CAD were evaluated using CACS. Blood samples were collected and stored at −80 °C for analysis of CTRP5 and chemerin levels via ELISA. **Results**: Serum CTRP5 and chemerin levels were significantly higher in the CAD group compared to the non-CAD group (221.83 ± 103.81 vs. 149.35 ± 50.99 ng/mL, *p* = 0.003 and 105.02 ± 35.62 vs. 86.07 ± 19.47 ng/mL, *p* = 0.005, respectively). Receiver operating characteristic (ROC) analysis showed that a CTRP5 cutoff of 172.30 ng/mL had 70% sensitivity and 73% specificity for identifying CAD, while a chemerin cutoff of 90.46 ng/mL had 61% sensitivity and 62% specificity. A strong positive correlation was observed between CTRP5 and chemerin, but neither adipokine showed a correlation with the Agatston score, a measure of CAD severity and extent, nor with coronary artery stenosis as determined by CCTA. **Conclusions**: CTRP5 and chemerin were significantly elevated in the CCTA-confirmed CAD group compared to the non-CAD group, with CTRP5 showing greater sensitivity and specificity. However, neither adipokine was linked to CAD severity and extent, differing from findings based on invasive coronary angiography (ICA). CTRP5 may serve as a promising “all-or-none biomarker” for CAD presence.

## 1. Introduction

Cardiovascular disease (CVD) is the predominant cause of mortality and morbidity worldwide [1]. CVD is responsible for over 39 million deaths annually in Europe, or 45% of total mortality. The most prevalent causes of these fatalities are coronary artery disease (CAD) and stroke, with CAD continuing to be the primary cause of mortality in Europe despite significant progress in treatment and prevention over the last thirty years. The worldwide incidence of CAD is rising and is projected to reach 1845 cases per 100,000 individuals by 2030 [2,3]. Coronary computed tomography angiography (CCTA) is now growing in its utility as it can detect both obstructive coronary artery disease and high-risk features of atherosclerotic plaques that cause stenosis, leading to an increased risk of ischemic events [4,5].

Biomarkers, such as adipokines, interleukins, and C-reactive proteins, have been investigated in several investigations and demonstrate potential for the early detection of CAD [6,7,8,9,10]. Early detection of CAD through biomarkers and CCTA is crucial for patient management, as it can help prevent ischemic events by optimizing pharmacological therapy or facilitating coronary interventional procedures [1,5].

Adipose tissue functions as an endocrine organ, producing various physiologically active peptides known as adipokines. These adipokines exhibit autocrine, paracrine, and endocrine roles, regulating critical processes such as adipose tissue metabolism, differentiation, energy equilibrium, and overall physiological homeostasis. Adipokines are crucial for sustaining metabolic balance and have been demonstrated to affect various biological processes, including immunological response, lipid metabolism, insulin sensitivity, vascular homeostasis, and angiogenesis. Consequently, adipokines may contribute to the pathophysiology of CVD in both direct and indirect pathways [11,12]. By influencing atherosclerosis through adipokine production, epicardial adipose tissue (EAT) close to coronary arteries raises the risk of cardiometabolic diseases [13,14,15]. The development of CAD is linked to the C1q/TNF-related protein (CTRP) family, particularly CTRP5, as well as other adipokines, such as chemerin. CTRP5 is a recently identified pro-atherogenic cytokine that promotes the transcytosis and oxidation of LDL within endothelial cells [16,17]. Despite research indicating a correlation between CTRP5 and atherosclerosis development, some investigations have documented reduced serum levels of CTRP5 in patients with CAD [18].

Chemerin is an adipokine that plays a significant role in the progression of metabolic diseases and inflammation-related disorders affecting the cardiovascular system. Chemerin modulates energy metabolism, adipogenesis, and angiogenesis. A positive link exists between CAD and serum chemerin levels, with chemerin levels also correlating with the severity of coronary lesions [19,20,21]. Another study revealed higher chemerin levels as an independent predictor of CAD. Additionally, plasma chemerin levels were reported to increase in patients with CAD and were related to an elevated risk of substantial unfavorable cardiac events in this group [22]. The release of chemerin in perivascular tissue has been shown to positively correlate with the progression of aortic and coronary atherosclerosis [23]. There is a correlation between chemerin and peripheral arterial stiffness [24].

This study sought to assess the possible significance of the adipokine CTRP5, which has been inadequately explored in relation to CAD and the adipokine chemerin in differentiating between individuals with CCTA-confirmed normal coronary arteries and patients with CCTA-confirmed CAD. The secondary purpose was to examine the correlation between the severity and extent of CAD, as assessed by the coronary artery calcium score (CACS), and serum levels of CTRP5 and chemerin in the CCTA-confirmed CAD group.

## 2. Materials and Methods

A total of 106 consecutive individuals who presented with chest pain were enrolled for this pilot trial between February 2024 and June 2024. These individuals were then prospectively evaluated utilizing CCTA. Atrial fibrillation, being younger than 35 years old, having a history of using lipid-lowering medications such as fibrates and statins, having chronic renal disease of stage 3 or greater, having a chronic liver disease, having an autoimmune disease, having been previously diagnosed with CAD or peripheral artery disease were all criteria that were used to exclude participants from the study.

In total, there were 106 individuals, and out of those, 89 were determined to be suitable for participation in the study. The CCTA-confirmed CAD group was defined as participants who had any soft, mixed, or calcific atherosclerotic plaque in their coronary arteries, whether or not coronary artery calcification (CAC) was present on anatomical examination by CCTA. The CACS, assessed by the Agatston score, was used as an objective measure of the extent and severity of CAD and was used to quantify the extent and severity of CAD in the CCTA-confirmed CAD group. Patients with zero CACS but any atherosclerotic plaque detected by CCTA were also included in the CCTA-confirmed CAD group. Participants without atherosclerotic plaque in their coronary arteries, as assessed by CCTA, and exhibiting zero CACS along with no signs of myocardial ischemia on treadmill/stress electrocardiogram or myocardial perfusion scintigraphy—conducted to rule out microvascular disease—were defined as the CCTA-confirmed non-CAD group (true normal coronary artery group). An illustration of the flow chart for the study can be observed in Figure 1.

A 128-slice single-source scanner (Somatom Go Top; Siemens Healthcare, Forchheim, Germany) was used for the procedure, and an independent expert was responsible for the evaluation of the results. Using the same CT scanner, the Agatston score was used to determine the degree of CACS.

Obtaining the patient’s consent allowed for the collection of clinical and laboratory data, which included information about the patient’s gender, age, height, weight, blood pressure measures, blood cholesterol levels, smoking history, and familial history of CAD. Blood samples were collected from the antecubital vein in the early morning hours of the morning, only after the patient had been fasting for at least eight hours. Following that, a biochemical analyzer was utilized in order to quantify serum lipids in addition to a number of other biochemical parameters. A fasting period of one night was followed by the collection of blood samples for CTRP5 and chemerin, which were then preserved at a temperature of −80 degrees Celsius. Quantification of serum levels of CTRP5 and chemerin was performed with the use of commercially available ELISA kits (Invitrogen ELISA Human C1qTNF5 ELISA Kit and Invitrogen Human RARRES2/TIG2 ELISA Kit) at appropriate dilutions in accordance with the directions provided by the manufacturer.

The protocol for the study was approved by the Bakircay University Non-Interventional Clinical Research Ethics Committee (Approved 24 August 2023; Decision number: 1154; Research number: 1155). All individuals who took part in the study provided written informed consent.

### Statistical Analysis

Continuous variables were presented as the mean ± standard deviation (SD), and the independent samples *t*-test was used to examine the data. The most relevant chi-squared test was utilized in order to perform the analysis on the categorical variables, which were represented as frequencies and percentages. When calculating the correlations between CTRP5, chemerin, and other variables, the Pearson correlation analysis was utilized as the method of evaluation. In order to diagnose CAD that was confirmed by CCTA, receiver operating characteristic (ROC) curves were established for CTRP5 and chemerin. The Statistical Package for the Social Sciences (SPSS), version 29.0 (SPSS Inc., Chicago, IL, USA) was utilized for each and every statistical analysis that was carried out. A two-tailed *t*-test was utilised, with a *p* value < 0.05 considered statistically significant.

## 3. Results

Demographic and clinical characteristics of the participants in both study groups are given in Table 1. No statistically significant difference was observed between the two groups regarding demographic risk factors related to the etiology of CAD (including age and gender), anthropometric measurements (such as height, weight, and body mass index), or biochemical and clinical parameters associated with CAD risk (including hypertension, diabetes, lipid profiles, and blood glucose levels).

The serum levels of the adipokines CTRP5 and chemerin exhibited significant differences between the study groups (Table 1). Both adipokines were significantly increased in the group with CCTA-confirmed CAD compared to the group with CCTA-confirmed non-CAD (*p* < 0.05).

The results of the ROC analysis presented in Figure 2 show that a cut-off value of 172.30 ng/mL was determined for CTRP5, which had a sensitivity of 70% and a specificity of 73% when it came to diagnosing CAD that was confirmed by CCTA. According to the results of our ROC analysis for chemerin, the sensitivity and specificity for identifying CCTA-confirmed CAD at a cut-off value of 90.46 ng/mL were found to be 61% and 62%, respectively (Figure 3).

According to the findings of our research, there is an important positive correlation between the levels of CTRP5 in plasma and the levels of chemerin in plasma. On the other hand, there was no correlation found between the plasma concentrations of either adipokine and the Agatston score (CACS), which is a measurement of the atherosclerotic burden and the severity of CAD (Table 2, Figure 4). Additionally, there was no association found between the amount of coronary artery stenosis that was indicated by CCTA and the levels of CTRP5 and chemerin that were found in the plasma. There was no correlation found between the levels of CTRP5 and chemerin in the plasma and any of the demographic, clinical, or biochemical data that were taken in the investigation.

## 4. Discussion

This study sought to assess differences in plasma levels of the adipokines CTRP5 and chemerin between individuals with CAD, characterized by mixed, calcific, or soft plaques across different stages of CCTA-confirmed atherosclerosis, and those without CAD, confirmed by CCTA. Additionally, the study aimed to investigate the correlation between chemerin and CTRP5 with the CACS (Agatston score), which allows for the objective detection of the severity and extent of CAD in the CCTA-confirmed CAD group [4,25,26,27,28,29]. The findings of our research revealed that the levels of CTRP5 and chemerin were significantly higher in the CCTA-confirmed CAD group compared to the CCTA-confirmed non-CAD group. Our findings indicate that serum CTRP5 levels show better sensitivity and specificity for the diagnosis of CAD compared to serum chemerin levels. A significant positive correlation was observed between serum CTRP5 and chemerin levels. In the CCTA-confirmed CAD group, there was no correlation between the severity and extent of CAD, measured by the CACS (Agatston score), and the serum levels of either adipokine. No statistically significant association was found between serum CTRP5 or chemerin levels and the degree of coronary artery stenosis in the CCTA-confirmed CAD group.

The formation of atherosclerosis involves various lipids, immune cells, vascular smooth muscle cells (VSMCs), and adipokines. CTRP5, an adipokine released from adipose tissue, particularly from EAT, may affect the formation of atherosclerotic plaques by multiple mechanisms [12,13,14,15,16,17,30]. Li et al. observed that serum CTRP5 levels were significantly higher in patients with CAD compared to individuals with normal coronary arteries, showing a correlation with the number of affected arteries. The study demonstrated elevated CTRP5 expression in the endothelium of early-stage atherosclerotic lesions [17]. Another investigation demonstrated that CTRP5 increased the expression of MMP2, cyclin D1, and TNF-alpha in a dose-dependent manner in aortic smooth muscle cells (ASMCs), thereby activating the Notch1, TGF-beta, and hedgehog pathways. This facilitated VSMC proliferation, inflammation, and migration during the early stages of atherosclerosis [31]. Macrophages within the arterial wall uptake modified LDL, resulting in foam cell formation and increased inflammation. Li et al. proposed that CTRP5 regulates 12/15-LOX through STAT6 signaling, thereby increasing LDL absorption and oxidation in macrophages and endothelial cells [17,30,32,33,34].

Liu et al. conducted a study comparing a normal coronary artery group with CAD patients experiencing acute coronary syndrome (ACS). Their findings indicated that, in agreement with our study, CTRP5 levels were significantly elevated in the CAD group compared to the normal coronary artery group, and there was no correlation between the severity and extent of CAD, as assessed by the Gensini Score, and CTRP5 levels [35]. This study significantly contrasts with ours, which utilized the CCTA method, as it utilized invasive coronary angiography (ICA) for diagnosis, established CAD patients with ACS in the CAD cohort, and classified individuals with less than 50% coronary stenosis as part of the normal coronary artery group.

In vitro studies indicate that chemerin increases the formation of reactive oxygen species (ROS) and inflammation in human endothelial cells and VSMCs, explaining its contribution to vascular dysfunction. Chemerin promotes the pro-inflammatory NF-κB pathway and enhances endothelial inflammation by promoting monocyte-endothelial adhesion [19,34]. Chemerin’s role in atherosclerosis is related to its interaction with macrophages through the chemerin chemokine-like receptor 1 (CMKLR-1) [36]. Immunohistochemical data show a relationship between the expression of chemerin and CMKLR-1 in human arteries and periadventitial adipose tissue, as well as the severity of atherosclerosis [19,37,38]. Chemerin contributes to the early stages of atherosclerosis by decreasing cGMP synthesis and nitric oxide-induced vasodilation, enhancing endothelial cell proliferation and migration, and further stimulating angiogenesis [19,39,40,41,42].

This study found that plasma chemerin levels were elevated in patients with CAD compared to those without CAD. This observation is consistent with prior research that has demonstrated a connection between chemerin concentrations and cardiovascular diseases [19,20,21,22,43,44]. Conversely, our data indicated that there is no link between the severity and extent of CAD and plasma chemerin levels. This finding sharply contrasts with prior research that demonstrated a correlation between chemerin levels and the extent and severity of CAD, as assessed by the Gensini Score derived from ICA results or the number of stenosed coronary arteries [20,43,44]. The lack of correlation between CAD severity and chemerin plasma levels in our study may be related to the more accurate effectiveness of CCTA in illustrating the comprehensive burden of atherosclerosis and the existence of atherosclerotic plaques at different stages of the disease relative to the ICA employed for CAD classification in previous investigations [25,26]. There is evidence that chemerin is implicated in the pathogenesis of atherosclerosis, particularly in the early stages of the disease [19,34,36,37,38,39,40,41].

Invasive coronary angiography was employed to evaluate the existence, severity, or stent restenosis of CAD in the majority of academic studies investigating the correlation between atherosclerosis and CTRP5, chemerin, and other adipokines in the past. The cohort referred to as the CAD group in prior research employing the ICA methodology typically comprised individuals with coronary artery stenosis exceeding 50%. Conversely, patients with stenosis less than 50% were arbitrarily designated as the control group or included in the cohort that had no restenosis following stent placement [17,31,35,43,44]. However, patients exhibiting lower than 50% stenosis or lacking intraluminal stenosis due to atherosclerotic eccentric plaques may not adequately reflect a truly normal cohort free of coronary artery atherosclerosis, particularly in studies employing ICA lumenographic criteria. These earlier studies’ control groups contained CAD patients, which adds a level of difficulty to the interpretation of the CAD prediction. The use of ICA to assess the severity of CAD is also problematic because it ignores the existence of eccentric plaques that do not contribute to the load of intraluminal plaque [25,26,31,35,43,44].

The present study preferred CCTA, which offers superior accuracy compared to ICA in diagnosing and excluding CAD. Eccentric plaques that do not induce intraluminal narrowing, which is undetectable by ICA, together with minor plaques in the initial stages of atherosclerosis, can be identified by CCTA [4,25,26]. Consequently, in the study, the CCTA-confirmed CAD group, indicative of coronary artery disease, was established with improved precision, while the CCTA-confirmed non-CAD group, representing the true normal coronary artery cohort, was constructed as optimally as possible.

Coronary artery calcification is a defining characteristic of atherosclerosis and a significant contributor to the onset and advancement of CAD. CACS reflects the overall burden of coronary atherosclerosis. Sangiorgi et al. found that coronary calcium measurement effectively assesses atherosclerotic plaque presence and burden, though no strong predictive link was found between luminal narrowing and calcification, possibly due to remodeling [27,28,29]. The Denmark Heart Registry, using CCTA, showed that total plaque burden “CACS” is a key factor in cardiovascular risk, regardless of stenosis, with similar CACS levels indicating similar event risk, whether CAD is obstructive or not [45].

Contrary to previous studies that identified a correlation between CAD severity and extent, as measured by the Gensini Score, SYNTAX score, or the number of coronary arteries with critical stenosis, and serum levels of CTRP5 or chemerin derived from ICA method data [17,31,43,44], our research did not find a relationship between CACS, a marker of CAD severity and extent [27,28,29], and serum concentrations of CTRP5 and chemerin. Consistent with our findings, Szpakowicz et al. observed no statistically significant link between the SYNTAX Score, which reflects the extent and severity of the disease, and chemerin serum levels in stable CAD patients having percutaneous coronary intervention [46].

The contrasting results may be explained by the advantages of the CCTA method used for defining CAD, which is more effective in detecting early-stage minor and eccentrically located atherosclerotic plaques [25,26] with elevated levels of CTRP5 and chemerin, compared to the ICA method employed in previous studies for CAD diagnosis. The CCTA method employed in this study effectively identifies early-stage plaques in cases of low CAD extent and the absence of intraluminal stenosis. However, CACS, used for assessing disease extent and severity, shows limited effectiveness in detecting early-stage plaques [28,29]. Consequently, patients with CAD exhibiting varying disease severities and different levels of intraluminal stenosis may present comparable early-stage plaque loads and similar serum levels of CTRP5 and chemerin. The CCTA method allows the identification of patients exhibiting advanced atherosclerosis, as evidenced by increased CACS, significant calcific plaque burden, and lowered early-stage atherosclerotic plaque presence, resulting in reduced levels of CTRP5 and chemerin expression. Conversely, patients with CAD who display elevated levels of CTRP5 and chemerin, minimal CAD extent, low CACS, and significant early-stage atherosclerotic plaque burden (without intraluminal stenosis) may also be identified by CCTA. This may clarify why plasma levels of CTRP5 and chemerin do not correlate with the severity, extent, or CACS of CAD. The absence of a correlation between the severity and extent of CAD, as assessed by CACS, and the levels of CTRP and chemerin in our study may be attributed to the limitations of the CACS method in detecting non-calcified atherosclerotic plaques during the early stages of atherosclerosis, where elevated levels of adipokines are anticipated [17,25,26,31,43,44].

### Limitations

We acknowledge that there are certain limitations, such as the relatively small number of the cohort, and the findings are based on results from a single center. In the CCTA-confirmed CAD group, the CACS methodology employed to evaluate the severity and extent of CAD is incapable of identifying early-stage atherosclerotic plaques that lack calcification. Because of this, additional validation is needed. This preliminary investigation has the potential to pave the way for far more extensive research in the future.

## 5. Conclusions

CTRP5 and chemerin levels were significantly elevated in the CCTA-confirmed CAD group compared to the CCTA-confirmed non-CAD group. CTRP5 was more sensitive and specific than chemerin in identifying CAD. The plasma levels of CTRP5 and chemerin were not found to be associated with CAD severity and extent in the CCTA-confirmed CAD group, in contrast to the results of previous studies utilizing ICA. Adipokine CTRP5 may serve as a promising “all-or-none” biomarker for CAD presence.

## Figures and Tables

**Figure 1 diagnostics-15-00206-f001:**
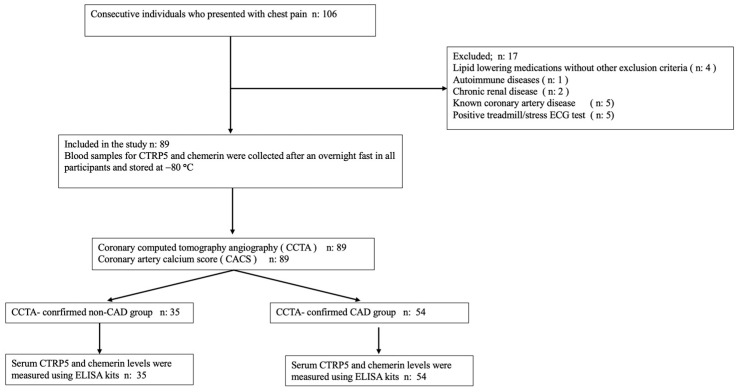
The study’s flowchart.

**Figure 2 diagnostics-15-00206-f002:**
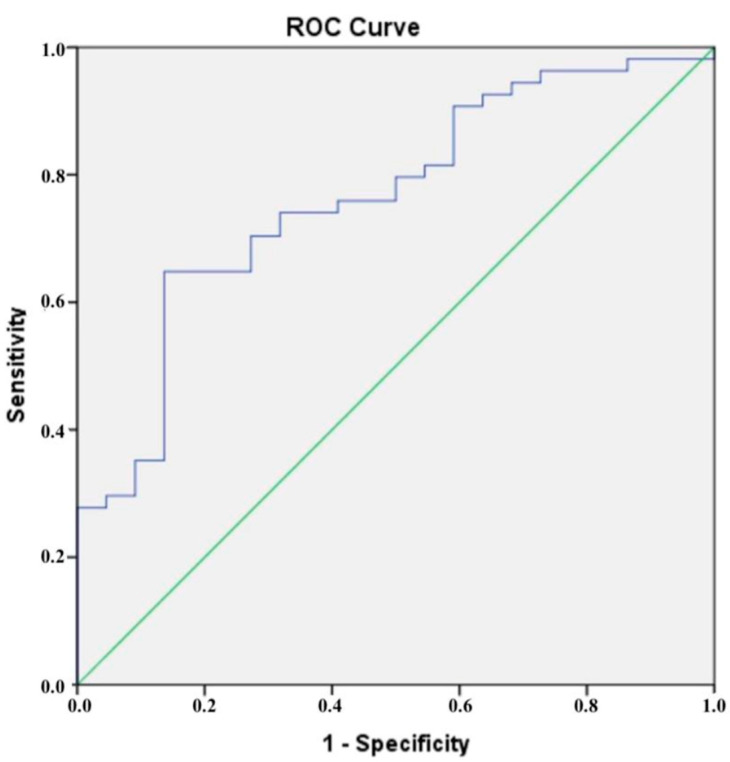
The blue line is the receiver operating characteristic (ROC) curve for CTRP.

**Figure 3 diagnostics-15-00206-f003:**
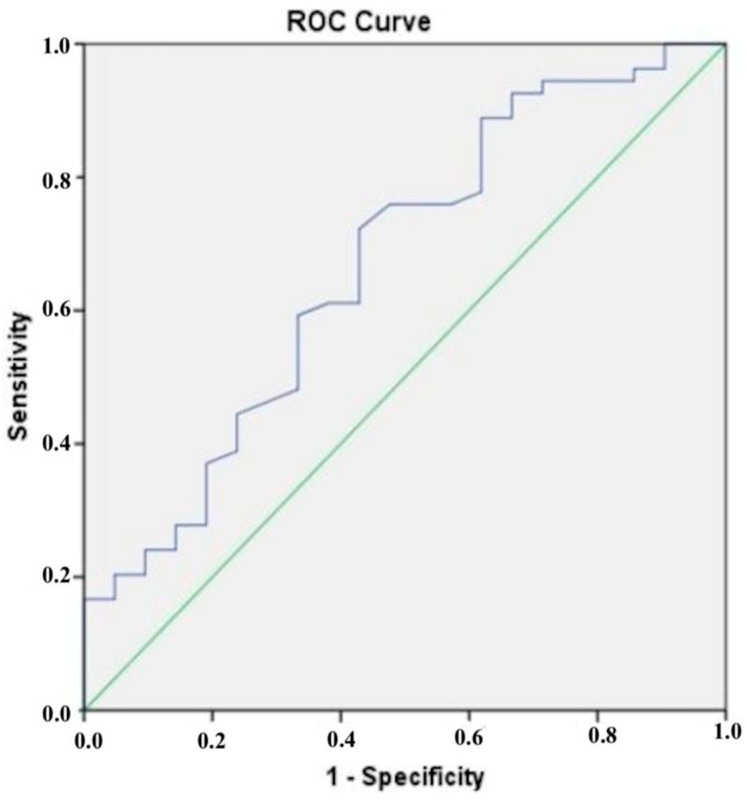
The blue line is the receiver operating characteristic (ROC) curve for chemerin.

**Figure 4 diagnostics-15-00206-f004:**
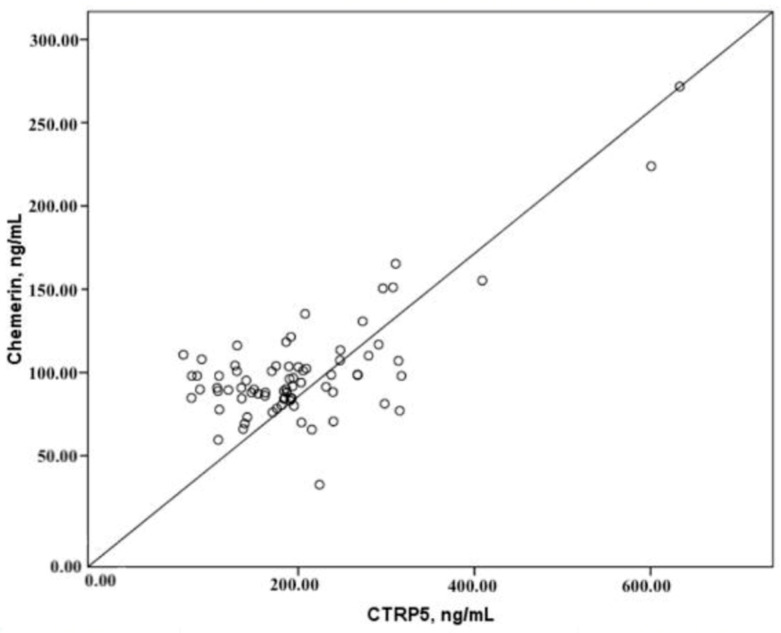
Correlations of CTRP5 and chemerin.

**Table 1 diagnostics-15-00206-t001:** Baseline demographic and clinical characteristics of participants.

Characteristics	CCTA-Confirmed Non-CAD Group *n* = 35	CCTA-Confirmed CAD Group *n* = 54	*p*
Age (years)	58.9 ± 7.7	56.7 ± 7.2	0.231
Gender (female %)	40	38.9	0.861
Smoking %	34	37	0.756
DM %	20	22.2	0.843
HT %	28.6	31.4	0.703
BMI (kg/m^2^)	28.1 ± 3.4	27.6 ± 3.1	0.535
Metabolic Syndrome %	25.7	27.7	0.756
Agatston score	0	224.0 ± 342.3	<0.001
Systolic Blood Pressure (mmHg)	135 ± 13	132 ± 25	0.623
Diastolic Blood Pressure (mmHg)	80 ± 10	79 ± 12	0.938
Fasting Blood Glucose (mg/dL)	100.9 ± 42.7	107.1 ± 35.0	0.514
Total Cholesterol (mg/dL)	225.7 ± 53.4	223.3 ± 52.3	0.858
Triglyceride (mg/dL)	153.4 ± 72.5	192.0 ± 116.0	0.083
LDL Cholesterol (mg/dL)	149.3 ± 38.1	140.0 ± 48.5	0.417
HDL Cholesterol (mg/dL)	50.0 ± 10.6	48.5 ± 11.9	0.592
Creatinine (mg/dL)	0.91 ± 0.04	0.93 ± 0.03	0.875
CTRP5 (ng/mL)	149.4 ± 51.0	221.8 ± 103.8	0.003
Chemerin (ng/mL)	86.1 ± 19.5	105.0 ± 35.6	0.005

**Table 2 diagnostics-15-00206-t002:** Correlations of CTRP5, chemerin, and Agatston score.

		Chemerin	CTRP5	Agatson
Chemerin	r		0.725 *	0.059
	*p*		0.000	0.613
CTRP5	r	0.725 *		0.061
	*p*	0.000		0.601
Agatson	r	0.059	0.061	
	*p*	0.613	0.601	

* *p* < 0.05.

## Data Availability

Biochemical data and computed tomographic coronary angiography images of all participants in our study are accessible in the electronic archive of our clinic. Furthermore, SPSS files created for statistical analysis are also documented. All data are accessible. If the journal editor desires, it may be shared with him/her.

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
