# Peer review of "Diagnostic Potential of CTRP5 and Chemerin for Coronary Artery Disease: A Study by Coronary Computed Tomography Angiography"

_diagnostics, 2025, doi:10.3390/diagnostics15020206_

Round 1
Reviewer 1 Report
Comments and Suggestions for Authors
Review: diagnostics-3348634
The manuscript entitled: ‘Diagnostic Potential of CTRP5 and Chemerin for Coronary Artery Disease: A Study by Coronary Computed Tomography Angiography’ is a well-structured and relevant study addressing the potential utility of CTRP5 and Chemerin in differentiating CAD from non-CAD patients using CCTA.
Despite its merit various aspects of the manuscript need clarification, refinement, and additional discussion.
Major revisions:
- The manuscript does not fully address the rationale behind the sequence of CCTA utilization and subsequent biomarker analysis. It would strengthen the study to clarify the choice of CCTA as the primary diagnostic tool over ICA, considering its limitations in detecting early-stage plaques without calcification.
- The absence of a correlation between CTRP5/Chemerin levels and CAD severity is discussed, but alternative explanations or hypotheses (e.g., the limitations of CACS for non-calcified plaques) could be elaborated.
- Did you perform a sub-analysis to investigate if there is there any correlation between the chemerin and CTRP5 and the plaque's density [ High density (>350 HU) - calcific plaques - Lower density (30–130 HU)- noncalcified plaques] and composition based on Hounsfield units (HU) in CTCA?
- The sensitivity and specificity cut-offs for CTRP5 and Chemerin, though statistically presented, need biological or clinical justification. Why were these thresholds chosen, and how do they compare to established biomarker cut-offs in other cardiovascular studies?
- The limitations of the study should clearly be stated. Mention the small sample size explicitly and its implications for generalizability.
Minor Revisions:
- Please highlight in the baseline demographic data table the significant parameters that may predispose to CAD or alter biomarker levels.
- Please be sure that all the figures and tables should be labelled clearly with comprehensive legends to aid interpretability.
- In Line 81. Why did you exclude patients with atrial fibrillation?
- In Line 105. What was the dilution of the samples. Was the dilution the same in the CAD group and in the non-CAD group?
- In Line 109. What do you mean by stating this date?
- Please try to use consistent terminology for biomarkers (e.g., refer to CTRP5 as "biomarker" or "adipokine" uniformly).
- Some key references are missing, particularly recent studies on coronary artery calcification [PMID: 39518492], CTRP5 [PMID: 32202952], and the role of Chemerin atherosclerosis [PMID: 39335646]. Ensure all citations are up-to-date and relevant.
Author Response
Major Revisions:
Comments: The manuscript does not fully address the rationale behind the sequence of CCTA utilization and subsequent biomarker analysis. It would strengthen the study to clarify the choice of CCTA as the primary diagnostic tool over ICA, considering its limitations in detecting early-stage plaques without calcification
Responds: Dear Reviewer;
In the study, we used CCTA, a method that is much more sensitive than ICA for the diagnosis of CAD. Because eccentric plaques and minor plaques in the early stages of atherosclerosis, which are not detected by ICA due to not causing intraluminal narrowing, can be detected by CCTA, we aimed to create the closest possible true normal coronary group (CCTA confirmed non-CAD group) and the CCTA confirmed CAD group, which can be very precisely identified including the early stages of atherosclerosis, using CCTA. CCTA, whether or not there is calcification, is a very sensitive and specific method for detecting atherosclerosis. In the study, we attempted to eliminate the handicap of subjective assessments like GENSINI and SYNTAX by using CACS (Agatston score), which can objectively measure the severity and extent of CAD in the CCTA confirmed CAD group where CAD was determined by CCTA. Due to your very important suggestion, we added a section explaining the topic at the beginning of the discussion section. The relevant text is below.
4. Discussion
This study aimed to assess differences in plasma levels of the adipokines CTRP5 and chemerin between individuals with CAD, characterized by mixed, calcific, or soft plaques across different stages of CCTA-confirmed atherosclerosis, and those without CAD, confirmed by CCTA. The study preferred CCTA, which offers superior accuracy compared to ICA in diagnosing and excluding CAD. Eccentric plaques that do not induce intraluminal narrowing, which are undetectable by ICA, together with minor plaques in the initial stages of atherosclerosis, can be identified by CCTA [4,7]. Consequently, in the study, the CCTA-Confirmed CAD group, indicative of coronary artery disease, was established with improved precision, while the CCTA confirmed Non-CAD group, representing the true normal coronary artery cohort, was constructed as optimally as possible. Additionally, the study aimed to investigate the correlation between chemerin and CTRP5 with the Agatston score (CACS), which allows for the objective detection of the severity and extent of CAD in the CCTA confirmed CAD group [4,24,25].
Comments: The absence of a correlation between CTRP5/Chemerin levels and CAD severity is discussed, but alternative explanations or hypotheses (e.g., the limitations of CACS for non-calcified plaques) could be elaborated.
Responds: Dear Reviewer; We have added the clever hypothesis you suggested regarding the inconsistency of CTRP5 and Chemerin with the severity and extent of CAD determined by CACS to the end of our discussion section, covering both adipokines. We also added this hypothesis to the limitations section. The relevant section is provided below.
Another explanation for the lack of association between the severity and extent of CAD determined by CACS and the levels of CTRP and chemerin in our study is the limitation in the CACS method in identifying non-calcified atherosclerotic plaques in the early stages of atherosclerosis, where adipokines are theoretically expected to be at higher levels [17,28,31,32,,41,42].
Comments: Did you perform a sub-analysis to investigate if there is there any correlation between the chemerin and CTRP5 and the plaque's density [ High density (>350 HU) - calcific plaques - Lower density (30–130 HU)- noncalcified plaques] and composition based on Hounsfield units (HU) in CTCA?
Responds: Dear Reviewer, unfortunately, we did not perform a statistical study of the correlation between plaque density and adipokine levels.
Comments:The sensitivity and specificity cut-offs for CTRP5 and Chemerin, though statistically presented, need biological or clinical justification. Why were these thresholds chosen, and how do they compare to established biomarker cut-offs in other cardiovascular studies?
Responds: Dear Reviewer
Adipokines CTRP5 and chemerin are experimental biomarkers. In the literature, there is no consensus on a cut-off value or normal value for the diagnosis of CAD. The cut-off values we determined in our study are the statistical results we calculated as the most optimal cut-off values based on the ROC analysis of the data obtained from our study, according to the presence or absence of CAD.
Comments: The limitations of the study should clearly be stated. Mention the small sample size explicitly and its implications for generalizability.
Responds: Dear Reviewer, the limitations section has been revised as per your recommendation, emphasizing the study's small sample size, its single-center design, and the necessity for validation of our findings. The final version of the relevant part is presented below.
Limitations: We acknowledge that there are certain limitations, such as the relatively small number of the cohort, and the findings are based on results from a single center. In the CCTA Confirmed CAD group, the CACS methodology employed to evaluate the severity and extent of CAD is incapable of identifying early-stage atherosclerotic plaques that lack calcification. Because of this, additional validation is needed. This preliminary investigation has the potential to pave the way for far more extensive research in the future.
Minor Revisions
Comments: Please highlight in the baseline demographic data table the significant parameters that may predispose to CAD or alter biomarker levels.
Responds: Dear Reviewer;
We have added the following revision to the results section based on your valuable suggestion. In Table 1, we more clearly indicated that there is no difference between the CAD patient and control groups in terms of risk factors, including demographic, anthropometric, clinical, and biochemical parameters that could predispose to CAD.
No statistically significant difference was observed between the two groups regarding demographic risk factors related to the etiology of CAD (including age and gender), anthropometric measurements (such as height, weight, and body mass index), or biochemical and clinical parameters associated with CAD risk (including hypertension, diabetes, lipid profiles, and blood glucose levels).
Comments: Please be sure that all the figures and tables should be labelled clearly with comprehensive legends to aid interpretability.
Responds: Dear Reviewer; we have checked the titles and labels of all the figures and tables in the article. We did not detect any issues, but if you have any additional suggestions, we would be happy to make the necessary revisions.
Comments: In Line 81. Why did you exclude patients with atrial fibrillation?
Responds: Dear Reviewer; although the advancements in the scanning speeds of computed tomography devices and imaging technologies today have allowed for the evaluation of coronary anatomy with CCTA in patients with Atrial Fibrillation, patients with high heart rate (HR) and HR variability have traditionally been excluded from CCTA research trials, and this non-invasive method has not been applied to patients with atrial fibrillation (AF) and/or HR over 65 bpm during screening. We excluded patients with atrial fibrillation from our study in order to achieve a more accurate visualization of coronary anatomy.
Comments: In Line 105. What was the dilution of the samples. Was the dilution the same in the CAD group and in the non-CAD group?
Responds: Dear Reviewer; all participants’ serums from the CCTA Confirmed CAD group and the CCTA Confirmed Non-CAD group included in our study were processed by the same technician, who was unaware of which group the serum belonged to, using the same company's ELISA kits and the same dilution ratio as defined by the company. All serums were diluted to the defined ratio with the standard dilution solution sold by the company along with the ELISA Kit.
Comments: In Line 109. What do you mean by stating this date?
Responds: Dear Reviewer; the date is the ethics committee approval date of the study. With your valid suggestion, we have rewritten the relevant section as follows to eliminate the ambiguity. Thank you very much for your suggestion.
The protocol for the study was approved by the Bakircay University Non-Interventional Clinical Research Ethics Committee (Approved 24th August 2023; Decision number: 1154; Research number: 1155). All individuals who took part in the study provided written informed consent.
Comments: Please try to use consistent terminology for biomarkers (e.g., refer to CTRP5 as "biomarker" or "adipokine" uniformly).
Responds:Dear reviewer, for CTRP5 and chemerin, we uniformly used the term "adipokine" throughout the article.
Comments: Some key references are missing, particularly recent studies on coronary artery calcification [PMID: 39518492], CTRP5 [PMID: 32202952], and the role of Chemerin atherosclerosis [PMID: 39335646]. Ensure all citations are up-to-date and relevant.
Responds: Dear Reviewer; I sincerely thank you for ensuring that our article references the new medical literature. The three references you suggested and the texts related to them have been added to our article.
The introduction section has been added with the text " Despite research indicating a correlation between CTRP5 and atherosclerosis development, some investigations have documented reduced serum levels of CTRP5 in patients with CAD [18]." and the following article has been referenced.
18) Moradi, N.; Fadaei, R.; Rashidbeygi, E.; Bagheri Kargasheh, F.; Malek, M.; Shokoohi Nahrkhalaji, A.; Fallah, S. Evaluation of changing the pattern of CTRP5 and inflammatory markers levels in patients with coronary artery disease and type 2 diabetes mellitus. Arch Physiol Biochem. 2022;128:964-969.
The following references have been added as sources 27 and 39 to the discussion section, and revisions have been made in the discussion section regarding the relevant references.
27) Mitsis, A.; Khattab, E.; Christodoulou, E.; Myrianthopoulos, K.; Myrianthefs, M.; Tzikas, S.; Ziakas, A.; Fragakis, N.; Kassimis, G. From Cells to Plaques: The Molecular Pathways of Coronary Artery Calcification and Disease. J Clin Med. 2024 ;13:6352.
39) Mitsis, A.; Khattab, E.; Myrianthefs, M.; Tzikas, S.; Kadoglou, N.P.E.; Fragakis, N.; Ziakas, A.; Kassimis, G. Chemerin in the Spotlight: Revealing Its Multifaceted Role in Acute Myocardial Infarction. Biomedicines. 2024, 12: 2133.

Reviewer 2 Report
Comments and Suggestions for Authors
It is with great interest that I read the article entitled “Diagnostic potential of CTRP5 and chemerin for coronary artery disease: a study by coronary computed tomography angiography”. The holy grail of coronary disease would be to find a biomarker of coronary artery disease that may predict events, however, we haven't been so lucky yet!
My major comments about this paper are:
1. My main concern is that there is confusion through the manuscript as to whether you are using the calcium score alone for this analysis, or whether you have done more than that. How have you defined your CCTA non-CAD group, for example? For lines 110-115: looking at this section, the presence of CACS, whilst it is a marker of coronary disease presence, does not exclude the presence of coronary artery disease. And hence, did all of your patients undergo both calcium scoring and also an anatomical assessment, and was it only people who had a positive calcium score who was included in the CAD group? I.e. what happened to the people who had coronary plaque present but a calcium score of 0? Because on lines 201-204: this is very different to what you have portrayed in the paper up until this point. You start out with saying that the study is based on CACS, but then are referring to mixed, calcific, or soft plaque, and I cannot see any specific results related to this in your results, and lines 324-328 in the limitation specifically say CACS. I think you are really comparing your assays with coronary calcification – whilst this is associated with coronary artery disease, this is not a definitive marker based on CCTA utility. Your paper needs to be carefully reframed with the above discrepancies considered.
2. I have also found that the discussion section generally truly does not discuss the results of the paper: there are lots of relevant studies mentioned, however, how they relate to the findings is not apparent. The key specific findings of the study need to be discussed through the discussion, and paralleled to the existing literature. This is only done at the very end of the discussion (lines 299 onwards are the best).
My minor comments:
1. ICA abbreviation in the abstract needs to be expanded (line 30)
2. Line 42: I think it’s difficult to say that CCTA is ‘essential’ at this point in time: whilst I agree that it has a role in detecting both obstructive disease and high risk features of the disease, we still do not have good evidence of using this technology repeated or in addition to invasive coronary angiography when we have an established diagnosis of CAD. I would suggest tempering this statement as (or something like ‘… CCTA is now growing in its utility as it detects both obstructive disease and associated high-risk lesion features.’
3. Line 44: again, the biomarkers listed do not truly aid in the early diagnosis of CAD - this needs to be reframed
4. Please consistently either capitalise or not chemerin (I don’t think it needs to be capitalised, but if you choose to that's ok)
5. Line 80: take out ‘at some point’ – you just need to state that people were recruited between a time period.
6. Line 84: it should be ‘… or having been previously diagnosed with CAD or peripheral artery disease….’ (not …having a diagnosed CAD or peripheral…’: this does not make sense)
7. I am not sure what the relevance is of including lines 85-89 (Individuals who exhibited positive results…’) given how the methods are currently written: either this needs to be clarified/simplified to saying something like ‘patients who presented with chest pain and who were deemed by their treating physician that CCTA was the next appropriate step in their management were included in this study.’ (or something similar), or, if you have a particular pathway that is followed (e.g. patients who present with chest pain in the centre undergo a functional assessment first, and if this is negative then go onto CCTA) then this should come first, and not second.
8. Lines 106-109: clearly all ethics committees approve studies differently. I would suggest reframing the paragraph as ‘The protocol for the study was approved by the Bakircay University Non-Interventional Clinical Research Ethics Committee (Approved 24th August 2023; Decision number: 1154; Research number: 1155). All individuals who took part in the study provided written informed consent.’
9. Line 111: when it is written independent, does this mean someone who was not involved directly in the study, or independent from knowing the assay results?
10. Line 125: the sentence that starts on this line should read ‘A two-tailed t-test was utilised, with a P value <0.05 considered statistically significant.’ (this is much simpler and easier to read)
11. Lines 129-131: I would suggest restructuring this sentence slightly to “Demographic and clinical characteristics of the participants in both study groups are given in Table 1.” (or something similar please).
12. Lines 136-139: do you have any diagrams/tables in your paper establishing this? These should be referenced here, as you do not mention any specific numbers except for the P value.
13. Table 1: I would suggest BMI measurements/calculations are to 1 decimal place only, the Agaston score, lipid results, and fasting blood glucose results should at most be to one decimal place as this is how we may report it, and likewise systolic BP and diastolic BP should be rounded to whole numbers as this is how we report blood pressure (and the accuracy of this would otherwise be questioned).
14. Line 214: it is not usual to start a sentence with an acronym, and hence this should start with ‘Invasive coronary angiography…’
15. Line 286: there appears to be a missing reference after the acronym (CMKLR-1).
16. Line 323: it would be better to start the limitations section as ‘We acknowledge that there are certain limitations, such as…’ unless it is a specific requirement of the journal to denote this way.
17. Line 339: there is a redundant full stop (.) after C.A.
Author Response
Major Comments
Comments: My main concern is that there is confusion through the manuscript as to whether you are using the calcium score alone for this analysis, or whether you have done more than that. How have you defined your CCTA non-CAD group, for example? For lines 110-115: looking at this section, the presence of CACS, whilst it is a marker of coronary disease presence, does not exclude the presence of coronary artery disease. And hence, did all of your patients undergo both calcium scoring and also an anatomical assessment, and was it only people who had a positive calcium score who was included in the CAD group? I.e. what happened to the people who had coronary plaque present but a calcium score of 0? Because on lines 201-204: this is very different to what you have portrayed in the paper up until this point. You start out with saying that the study is based on CACS, but then are referring to mixed, calcific, or soft plaque, and I cannot see any specific results related to this in your results, and lines 324-328 in the limitation specifically say CACS. I think you are really comparing your assays with coronary calcification – whilst this is associated with coronary artery disease, this is not a definitive marker based on CCTA utility. Your paper needs to be carefully reframed with the above discrepancies considered.
Responds: Dear Reviewer,
In our investigation, the assessment of all patients for coronary artery disease was performed utilizing computed coronary tomography angiography (CCTA). In the anatomical assessment performed using CCTA, patients exhibiting atherosclerotic plaques in their coronary arteries were categorized into the CAD group, hence constituting the CCTA confirmed CAD group. Participants free of atherosclerotic plaques in their coronary arteries as determined by CCTA were classified as the truly normal coronary artery group, constituting the CCTA Confirmed Non-CAD group. The associations of CTRP5 and Chemerin with the prediction of CAD were established based on the presence or absence of plaques in CCTA, rather than CACS.
The CACS (Agatston score) was not utilized in the diagnosis or exclusion of coronary artery disease (CAD). All patients received CCTA, and CACS was routinely performed during the procedure for both the "CCTA-Confirmed CAD" group and the "CCTA Confirmed Non-CAD" group. The CACS (Agatston Score) has been employed to examine the correlation between disease severity and extent with chemerin and CTRP5 in the CCTA Confirmed CAD cohort, as it provides an objective and precise assessment of disease prevalence and severity. The lack of CACS does not exclude the existence of soft non-calcified atherosclerotic plaques, as you have accurately explained. We have revised the sections in the methods and discussion parts where your concerns were addressed. The relevant parts are provided below. (Bold and underlined section)
As you mentioned, in the entire CCTA-Confirmed CAD group, the atherosclerotic plaques detected by CCTA were classified as soft, mixed, and calcified plaques. However, due to the small sample size and the difficulty in objectively determining the intensity of the plaques in each patient based on their characteristics, no specific analysis was conducted on the relationship between plaque types and adipokines.
In the Limitation section, specifically in lines 324-328 that you mentioned, we wanted to describe the limitation of CACS in determining the severity and extent of the disease in the patient group due to its inability to detect non-calcified soft plaques. This is consistent with your explanations. In our study, the use of CACS does not pose a limitation for the diagnosis or exclusion of CAD because the diagnosis and exclusion of CAD were performed using the CCTA method, not CACS
2. Materials and Methods
A total of 106 consecutive individuals who presented with chest pain were enrolled for this pilot trial between February 2024 and June 2024. These individuals were then prospectively evaluated utilizing CCTA. Atrial fibrillation, being younger than 35 years old, having a history of using lipid-lowering medications such as fibrates and statins, having chronic renal disease of stage 3 or greater, having a chronic liver disease, having an autoimmune disease, having been previously diagnosed with CAD or peripheral artery disease were all criteria that were used to exclude participants from the study.
In total, there were 106 individuals, and out of those, 89 were determined to be suitable for participation in the study. The CCTA-confirmed CAD group was defined as participants who had any soft, mixed or calcific atherosclerotic plaque in their coronary arteries, whether or not CAC was present on anatomical examination by CCTA. The coronary artery calcification score (CACS), assessed by the Agatston score, was used as an objective measure of the extent and severity of CAD and was used to quantify the extent and severity of CAD in CCTA Confirmed CAD group. Patients with zero CACS but atherosclerotic plaque detected by CCTA were also included in the CCTA confirmed CAD group. Participants without atherosclerotic plaque in their coronary arteries, as assessed by CCTA, and exhibiting zero CACS along with no signs of myocardial ischemia on treadmill/stress electrocardiogram or myocardial perfusion scintigraphy—conducted to rule out microvascular disease—were defined as the CCTA-confirmed non-CAD group (true normal coronary artery group). An illustration of the flow chart for the study could be observed in Figure 1.
Obtaining the patients' consent allowed for the collection of clinical and laboratory data, which included information about the patients' gender, age, height, weight, blood pressure measures, blood cholesterol levels, smoking history, and familial history of CAD. Blood samples were collected from the antecubital vein in the early morning hours of the morning, only after the patient had been fasting for at least eight hours. Following that, a biochemical analyzer was utilized in order to quantify serum lipids in addition to a number of other biochemical parameters. A fasting period of one night was followed by the collection of blood samples for CTRP5 and chemerin, which were then preserved at a temperature of -80 degrees Celsius. Quantification of serum levels of CTRP5 and chemerin was performed with the use of commercially available ELISA kits (Invitrogen ELISA Human C1qTNF5 ELISA Kit and Invitrogen Human RARRES2/TIG2 ELISA Kit) at appropriate dilutions in accordance with the directions provided by the manufacturer.
The protocol for the study was approved by the Bakircay University Non-Interventional Clinical Research Ethics Committee (Approved 24th August 2023; Decision number: 1154; Research number: 1155). All individuals who took part in the study provided written informed consent.
Each participant underwent CCTA to determine coronary artery anatomy and the presence or absence of atherosclerotic plaque (CAD) in the coronary arteries. A 128-slice single-source scanner (Somatom Go Top; Siemens Healthcare, Forchheim, Germany) was used for the procedure, and an independent expert was responsible for the evaluation of the results. Using the same CT scanner, the Agatston score was used to determine the degree of CACS.
4. Discussion
This study aimed to assess differences in plasma levels of the adipokines CTRP5 and chemerin between individuals with CAD, characterized by mixed, calcific, or soft plaques across different stages of CCTA-confirmed atherosclerosis, and those without CAD, confirmed by CCTA. The study preferred CCTA, which offers superior accuracy compared to ICA in diagnosing and excluding CAD. Eccentric plaques that do not induce intraluminal narrowing, which are undetectable by ICA, together with minor plaques in the initial stages of atherosclerosis, can be identified by CCTA [4,7]. Consequently, in the study, the CCTA-Confirmed CAD group, indicative of coronary artery disease, was established with improved precision, while the CCTA confirmed Non-CAD group, representing the true normal coronary artery cohort, was constructed as optimally as possible. Additionally, the study aimed to investigate the correlation between chemerin and CTRP5 with the Agatston score (CACS), which allows for the objective detection of the severity and extent of CAD in the CCTA confirmed CAD group [4,24,25].
Limitations: We acknowledge that there are certain limitations, such as the relatively small number of the cohort, and the findings are based on results from a single center. In the CCTA Confirmed CAD group, the CACS methodology employed to evaluate the severity and extent of CAD is incapable of identifying early-stage atherosclerotic plaques that lack calcification. Because of this, additional validation is needed. This preliminary investigation has the potential to pave the way for far more extensive research in the future.
Comments I have also found that the discussion section generally truly does not discuss the results of the paper: there are lots of relevant studies mentioned, however, how they relate to the findings is not apparent. The key specific findings of the study need to be discussed through the discussion, and paralleled to the existing literature. This is only done at the very end of the discussion (lines 299 onwards are the best).
Responds: Dear Reviewer; In the literature, there are very few studies that specifically examine the relationship between CTRP5 and CAD. As far as we know, our study is one of the pioneering studies conducted with CCTA. Therefore, it was possible to establish the relationship between the reference articles and our study using histochemical and basic biochemical data. With your suggestion, we more clearly emphasized that CTRP5 levels are high in the early stages of atherosclerosis in these studies. Based on this information, contrary to previous studies, we attempted to explain the lack of correlation between the prevalence and severity of CAD detected by ICA and CTRP5 levels in our study. Furthermore, we have relocated the underlined segment, which interrupts the continuity by causing a disjunction in the section detailing the function of CTRP5 in the pathogenesis of atherosclerosis, to the portion indicated in bold and italic. We believe that the text, in its current shape, has become easier to read and logical.
The findings indicate that CTRP5 enhances the proliferation, inflammation, and migration of VSMCs via the stimulation of many pathways of signaling in early stages of atherosclerosis [32]. Infiltrating macrophages in the artery wall then absorb this altered LDL, resulting in the production of foam cells and heightened inflammatory reactions. 12/15-LOX is crucial for facilitating the absorption and oxidation of LDL in macrophages [35-38]. In the previous investigation, Li et al. proposed a theory indicating that 12/15-LOX is modulated by CTRP5 via STAT6 signaling in early stages of atherosclerosis. Additionally, the researchers showed that CTRP5 increased transcytosis of LDL across endothelium mono layers and the oxidative alteration of LDL in the endothelial cells [17].
The CCTA-confirmed CAD patients in our study had significantly higher plasma CTRP5 levels compared to individuals in the CCTA-confirmed non-CAD group. CTRP5 levels, however, did not correlate with the amount of CACS or the severity of CAD. We believe this inconsistency arises from the difference between the CCTA method used in our study to diagnose CAD and the ICA method employed by Li et al. The CCTA method in our study is capable of detecting CAD in patients with soft or eccentric plaques in the early stages of atherosclerosis, which exhibit low CACS and do not cause significant intraluminal narrowing [25,26], despite high CTRP5 expression by endothelial cells during these early stages[17]. Additionally, CCTA can also detect patients with severe intraluminal stenosis or extensive CAD, which present with high CACS and plaques at various stages of atherosclerosis. This difference may help explain why plasma CTRP5 levels are not associated with CAD severity, extent, or CACS. CCTA allows for the identification of patients with varying severities and extents of CAD, but with similar early-stage atherosclerotic plaque burdens and CTRP5 expression. CCTA can identify patients with varying severities and extents of CAD, while having similar early-stage atherosclerotic plaque burdens and CTRP5 expression levels. CCTA allows the identification of patients with advanced atherosclerosis, marked by significant intraluminal stenosis or extensive CAD, exhibiting low CTRP5 expression and increased CACS. Conversely, it also identifies CAD patients with early-stage atherosclerotic plaque burden, characterized by high CTRP5 levels, who do not yet exhibit intraluminal stenosis, with low CAD extents and low CACS. This circumstance might clarify why plasma CTRP5 levels do not correlate with the severity, extent, or CACS of CAD.
Infiltrating macrophages in the artery wall then absorb this altered LDL, resulting in the production of foam cells and heightened inflammatory reactions [33-36]. 12/15-LOX is crucial for facilitating the absorption and oxidation of LDL in macrophages [37]. In the previous investigation, Li et al. proposed a theory indicating that 12/15-LOX is modulated by CTRP5 via STAT6 signaling in early stages of atherosclerosis. Additionally, the researchers showed that CTRP5 increased transcytosis of LDL across endothelium mono layers and the oxidative alteration of LDL in the endothelial cells [17].The study by Liu et al. conducted a comparison between the normal coronary artery group and patients with CAD experiencing acute coronary syndrome (ACS). Consistent with our study's findings, it was shown that the CAD group's CTRP5 levels were significantly elevated compared to those of the normal coronary artery group. This previous study's definition of CAD, which encompassed lesions with a diameter stenosis exceeding 50% in the ICA and included patients with ACS, distinguishes it from the subject of our study.
Minor Comments
Comments: ICA abbreviation in the abstract needs to be expanded (line 30)
Responds: Dear reviewer, It was done.
Comments: Line 42: I think it’s difficult to say that CCTA is ‘essential’ at this point in time: whilst I agree that it has a role in detecting both obstructive disease and high risk features of the disease, we still do not have good evidence of using this technology repeated or in addition to invasive coronary angiography when we have an established diagnosis of CAD. I would suggest tempering this statement as (or something like ‘… CCTA is now growing in its utility as it detects both obstructive disease and associated high-risk lesion features.’
Comments: Dear Reviewer; revised version:
Coronary computed tomography angiography (CCTA) is now growing in its utility as it can detect both obstructive coronary artery disease and high-risk features of atherosclerotic plaques that cause stenosis, leading to an increased risk of ischemic events [4,5].
Comments: Line 44: again, the biomarkers listed do not truly aid in the early diagnosis of CAD - this needs to be reframed
Responds: The section rightly criticized by the esteemed Reviewer has been reframed, softening the sentence:
Biomarkers, such adipokines, interleukins, and C-reactive protein, have been investigated in several investigations and demonstrate potential for the early detection of CAD [6-9].
Comments: Please consistently either capitalise or not chemerin (I don’t think it needs to be capitalised, but if you choose to that's ok)
Responds: We agree with the esteemed Reviewer and have corrected all instances of "chemerin" that do not require capitalization according to grammatical rules, as suggested by the Reviewer.
Comments: Line 80: take out ‘at some point’ – you just need to state that people were recruited between a time period.
Responds: The expression has been corrected in the manner suggested by the esteemed reviewer. The corrected version of the relevant section is below.
A total of 106 consecutive individuals who presented with chest pain were enrolled for this pilot trial between February 2024 and June 2024.
Comments: Line 84: it should be ‘… or having been previously diagnosed with CAD or peripheral artery disease….’ (not …having a diagnosed CAD or peripheral…’: this does not make sense)
Responds: The expression has been corrected in the manner suggested by the esteemed reviewer. The corrected version of the relevant section is below.
Atrial fibrillation, being younger than 35 years old, having a history of using lipid-lowering medications such as fibrates and statins, having chronic renal disease of stage 3 or greater, having a chronic liver disease, having an autoimmune disease or having been previously diagnosed with CAD or peripheral artery disease were all criteria that were used to exclude participants from the study.
Comments: I am not sure what the relevance is of including lines 85-89 (Individuals who exhibited positive results…’) given how the methods are currently written: either this needs to be clarified/simplified to saying something like ‘patients who presented with chest pain and who were deemed by their treating physician that CCTA was the next appropriate step in their management were included in this study.’ (or something similar), or, if you have a particular pathway that is followed (e.g. patients who present with chest pain in the centre undergo a functional assessment first, and if this is negative then go onto CCTA) then this should come first, and not second.
Responds: With the valuable suggestion of Reviewer, the part causing conceptual confusion in the text ( lines 85-89 ) was removed from the article. Additionally, following his suggestion, we added a section defining the CCTA Confirmed CAD group and the CCTA Confirmed Non-CAD group. To clarify the confusion, we added that patients with positive results on treadmill/stress electrocardiogram testing or myocardial perfusion scintigraphy were excluded from the control group to rule out microvascular coronary artery disease in the CCTA Confirmed Non-CAD group. The corrected version of the relevant section is below.
Individuals who exhibited positive results on treadmill/stress electrocardiogram testing or myocardial perfusion scintigraphy indicating myocardial ischemia were subsequently referred for direct invasive coronary angiography (ICA), in accordance with established local protocols and clinical guidelines. This was done in order to enable potential interventional treatments and to more precisely identify the normal coronary group so as to exclude the coronary artery disease group that had microvascular disease. In total, there were 106 individuals, and out of those, 89 were determined to be suitable for participation in the study. The CCTA-confirmed CAD group was defined as participants who had any soft, mixed or calcific atherosclerotic plaque in their coronary arteries, whether or not CAC was present on anatomical examination by CCTA. The CACS, assessed by the Agatston score, was used as an objective measure of the extent and severity of CAD and was used to quantify the extent and severity of CAD in CCTA Confirmed CAD group. Patients with zero CACS but atherosclerotic plaque detected by CCTA were also included in the CCTA confirmed CAD group. Participants who had no atherosclerotic plaque in their coronary arteries on evaluation with CCTA and who also had zero CACS and no treadmill/stress electrocardiogram testing or myocardial perfusion scintigraphy indicating myocardial ischemia aimed to exclude microvascular disease were defined as the CCTA-confirmed non-CAD group (true normal coronary artery group).
Comments: Lines 106-109: clearly all ethics committees approve studies differently. I would suggest reframing the paragraph as ‘The protocol for the study was approved by the Bakircay University Non-Interventional Clinical Research Ethics Committee (Approved 24thAugust 2023; Decision number: 1154; Research number: 1155). All individuals who took part in the study provided written informed consent.’
Responds: The relevant section (lines 106-109) was entirely removed from the text and revised as per the valuable reviewer's suggestions. The corrected version of the relevant section is below.
The protocol for the study was approved by the Bakircay University Non-Interventional Clinical Research Ethics Committee (Approved 24th August 2023; Decision number: 1154; Research number: 1155). All individuals who took part in the study provided written informed consent.
Comments: Line 111: when it is written independent, does this mean someone who was not involved directly in the study, or independent from knowing the assay results?
Responds: The term is used to describe a physician who is unaware of the analysis results and evaluates the CCTA examinations blindly without falling into bias.
Comments: Line 125: the sentence that starts on this line should read ‘A two-tailed t-test was utilised, with a P value <0.05 considered statistically significant.’ (this is much simpler and easier to read)
Responds: The relevant section (line 125) was entirely removed from the text and revised as per the valuable reviewer's suggestions. The corrected version of the relevant section is below.
The Statistical Package for the Social Sciences (SPSS), version 29.0 (SPSS Inc., Chicago, Illinois, United States of America) was utilized for each and every statistical analysis that was carried out. A two-tailed t-test was utilised, with a P value <0.05 considered statistically significant.
Comments: Lines 129-131: I would suggest restructuring this sentence slightly to “Demographic and clinical characteristics of the participants in both study groups are given in Table 1.” (or something similar please).
Responds: The relevant section (lines 129-131) was entirely removed from the text and revised as per the valuable reviewer's suggestions. The corrected version of the relevant section is below.
Demographic and clinical characteristics of the participants in both study groups are given in Table 1. There were no significant differences found between the two groups in terms of demographic factors (such as age and gender), anthropometric measurements (such as height, weight, and body mass index), or clinical and biochemical parameters (such as hypertension, diabetes, lipid profiles, and blood glucose levels). No statistically significant difference was observed between the two groups regarding demographic risk factors related to the etiology of CAD (including age and gender), anthropometric measurements (such as height, weight, and body mass index), or biochemical and clinical parameters associated with CAD risk (including hypertension, diabetes, lipid profiles, and blood glucose levels).
Comments: Lines 136-139: do you have any diagrams/tables in your paper establishing this? These should be referenced here, as you do not mention any specific numbers except for the P value.
Responds: Dear Reviewer; in the last two rows of Table 1, the average values and p-value of CTRP5 and chemerin in the CCTA Confirmed CAD and CCTA Confirmed Non CAD groups were provided. It is marked in red and bold in the table. The expression (table 1) was added to the text in accordance with your suggestion. The text and table are shown below.
The serum levels of the adipokines CTRP5 and chemerin exhibited significant differences between the study groups (table 1). Both adipokines were significantly increased in the group with CCTA-confirmed CAD compared to the group with CCTA-confirmed non-CAD (p < 0.05).
Table 1. Baseline Demographic and Clinical Characteristics of Participants
Characteristics |
CCTA-confirmed nonCAD group n= 35 |
CCTA-confirmed CAD group n= 54 |
P |
Age (years) |
58.9 ± 7.7 |
56.7 ± 7.2 |
0.231 |
Gender (female %) |
40 |
38.9 |
0.861 |
Smoking % |
34 |
37 |
0.756 |
DM % |
20 |
22.2 |
0.843 |
HT % |
28.6 |
31.4 |
0.703 |
BMI (kg/m2 ) |
28.1 ± 3.4 |
27.6 ± 3.1 |
0.535 |
Metabolic Syndrome % |
25.7 |
27.7 |
0.756 |
AGATSTON |
0 |
224.0 ± 342.3 |
<0.001 |
Systolic Blood Pressure (mmHg) |
135 ± 13 |
132 ± 25.0 |
0.623 |
Diastolic Blood Pressure (mmHg) |
80 ± 10 |
79 ± 12 |
0.938 |
Fasting Blood Glucose (mg/dL) |
100.9 ± 42.7 |
107.1 ± 35.0 |
0.514 |
Total Cholesterol(mg/dL) |
225.7 ± 53.4 |
223.3 ± 52.3 |
0.858 |
Triglyceride (mg/dL) |
153.4 ± 72.5 |
192.0 ± 116.0 |
0.083 |
LDL Cholesterol (mg/dL) |
149.3 ± 38.1 |
140.0 ± 48.5 |
0.417 |
HDL Cholesterol (mg/dL) |
50.0 ± 10.6 |
48.5 ± 11.9 |
0.592 |
Creatinine (mg/dl) |
0.91 ± 0.04 |
0.93 ± 0.03 |
0.875 |
CTRP5 (ng/mL) |
149.4±51.0 |
221.8±103.8 |
0.003 |
Chemerin (ng/mL) |
86.1±19.5 |
105.0±35.6 |
0.005 |
Comments: Table 1: I would suggest BMI measurements/calculations are to 1 decimal place only, the Agaston score, lipid results, and fasting blood glucose results should at most be to one decimal place as this is how we may report it, and likewise systolic BP and diastolic BP should be rounded to whole numbers as this is how we report blood pressure (and the accuracy of this would otherwise be questioned).
Responds: The requested rounding of numbers for Table 1, as suggested by the esteemed Reviewer, has been completed, and the numbers have been adjusted to the desired single decimal place. The final version of Table 1 is provided above
Comments: Line 214: it is not usual to start a sentence with an acronym, and hence this should start with ‘Invasive coronary angiography. …’
Responds: Dear reviewer; the corrected version of the relevant section is below.
Invasive coronary angiography was employed to evaluate the existence, severity, or stent restenosis of CAD in the majority of academic studies investigating the correlation between atherosclerosis and CTRP5, chemerin, and other adipokines in the past.
Comments: Line 286: there appears to be a missing reference after the acronym (CMKLR-1).
Responds: Dear reviewer; the corrected version of the relevant section is below. Reference 39 added after the acronym (CMKLR-1).
There is a strong connection between chemerin and atherosclerosis because of the impact that it has on macrophages through the chemerin chemokine-like receptor 1 (CMKLR-1) [39].
Comments: Line 323: it would be better to start the limitations section as ‘We acknowledge that there are certain limitations, such as…’ unless it is a specific requirement of the journal to denote this way.
Responds:
Dear Reviewer; the corrected version of the relevant section is below.
Limitations: We acknowledge that there are certain limitations, such as the relatively small number of the cohort, and the findings are based on results from a single center. In the CCTA Confirmed CAD group, the CACS methodology employed to evaluate the severity and extent of CAD is incapable of identifying early-stage atherosclerotic plaques that lack calcification. Because of this, additional validation is needed. This preliminary investigation has the potential to pave the way for far more extensive research in the future.
Comments: Line 339: there is a redundant full stop (.) after C.A.
Responds: Dear Reviewer; the corrected version of the relevant section is below.
Author Contributions: Conceptualization, T.O. and C.T.; methodology, T.O., C.A., C.T, M,B,Y; software, M.D., M.B.Y, T.O.; validation, T.O., M.B.Y.; formal analysis, M.D.; investigation, T.O., C.T., C.A.; resources, T.O.; data curation, C.T, T.O.,M.D.; Writing—initial draft preparation, T.O., M.B.Y.; writing—review and editing, M.B.Y.; visualization, T.O.; supervision, M.B.Y.; project administration, T.O. “All authors have read and agreed to the published version of the manuscript.”

Round 2
Reviewer 1 Report
Comments and Suggestions for Authors
All my comments have been addressed succesfully by the authors.
Author Response
Comments 1: All my comments have been addressed succesfully by the authors.
Response 1: Dear Reviewer, your insightful criticism and recommendations have significantly improved the quality and clarity of our article. I greatly appreciate your help and guidance.
Reviewer 2 Report
Comments and Suggestions for Authors
Thank you to the authors for your responses and for clarifying how your CCTA-confirmed CAD group was derived, as this is very helpful and provides significant clarity for the paper. However, and I apologise for not making this clearer during the first review, but it is not clear exactly what you are measuring against in the primary outcome when you compare the abstract and the main manuscript, which is why I brought up this point originally: specifically, in the abstract background, you refer to CCTA-confirmed CAD vs non-CAD, and then, following In the methods of the abstract, you write that CAD severity and extent were evaluated using CACS, and then divided into two groups (this statement makes me think that you are basing your CCTA confirmed CAD and non-CAD exclusively on a CACS). You have now written (or rather, clarified) in the methods section of the main paper that that CCTA-confirmed CAD group were defined as any plaque on CCTA, not just calcium (and in the discussion) – given that the latter seems to be correct, this needs to be clarified in your abstract to make your messaging consistent. You then mention in the discussion regarding comparing CACS and chemerin and CTRP5 – this is a secondary outcome? Consistency is required between the abstract and main manuscript.
My second main comment, and I’m sorry again that I did not make this clear in my initial review, but I understand that there is limited literature about CTRP5 and chemerin and its relationship with CCTA-diagnosed atherosclerosis (hence the need for your study), but reading through your discussion, not all of your key findings seem to be highlighted, and now with the additions of the additional lines in the paragraph, it does not flow well with what a reader would expect to read. The paragraph beginning on line 245 should essentially be your second paragraph, with the discussion regarding CTRP5 following, and then chemerin. Then a discussion about the lack of correlation with the Agaston score is required, and this is when/where you should include the paragraph beginning on line 223. You should consider:
- Lines 213-222: this is not what I expect to read at this point in the discussion. After your opening sentence, I expect to read about a summary of your key findings. What you have written currently belongs in other places around the discussion (particularly before your limitations, as you are essentially summarising your strengths); lines 220-222 may be able to follow-on form the first sentence.
- The paragraph beginning on line 231 should come first, and the line should read ‘In the past, invasive coronary angiography has been employed to evaluate the existence, severity, or stent restenosis of CAD in the majority of studies investigating the correlation between atherosclerosis and CTRP5, chemerin, and other adipokines. This paragraph should also be thought of as a strengths paragraph, and put towards the end of the discussion prior to the limitations
- The paragraph beginning on line 223 should come later in the discussion, and be related to your CACS findings
- The discussion also needs to be read carefully to ensure that there is minimal repetition from beginning to end as it is quite long
Minor issues still detected in the manuscript:
Lines 115-118 should be moved to the end of the methods section, but before the statistical analysis section.
Lines 119-120 (Each participant… in the coronary arteries.) is not needed as this is implied above.
Lines 120-124 should probably go at the end of the paragraph ending on line 102.
Systolic BP in table 1 in the CCTA confirmed CAD group: remove the .0 in the SD value.
Table 2. there are commas instead of full stops in the numbers, and all numbers should have leading zeros in them
Line 217 – CCTA-confirmed CAD group (confirmed needs a lower-case c and similarly non-CAD group on line 218 should be lower-case n (and this should be consistent through the paper)
Author Response
Major Comments
Comments 1: Thank you to the authors for your responses and for clarifying how your CCTA-confirmed CAD group was derived, as this is very helpful and provides significant clarity for the paper. However, and I apologise for not making this clearer during the first review, but it is not clear exactly what you are measuring against in the primary outcome when you compare the abstract and the main manuscript, which is why I brought up this point originally: specifically, in the abstract background, you refer to CCTA-confirmed CAD vs non-CAD, and then, following In the methods of the abstract, you write that CAD severity and extent were evaluated using CACS, and then divided into two groups (this statement makes me think that you are basing your CCTA confirmed CAD and non-CAD exclusively on a CACS). You have now written (or rather, clarified) in the methods section of the main paper that that CCTA-confirmed CAD group were defined as any plaque on CCTA, not just calcium (and in the discussion) – given that the latter seems to be correct, this needs to be clarified in your abstract to make your messaging consistent. You then mention in the discussion regarding comparing CACS and chemerin and CTRP5 – this is a secondary outcome? Consistency is required between the abstract and main manuscript.
Responses 1: Dear Reviewer; in the methods section of the first revision we made based on your suggestion, we have more clearly defined how the CCTA-confirmed CAD group and the CCTA-confirmed non-CAD groups were formed. Thanks to your suggestion, the writing gained a clearer and more fluent narrative. Thank you very much for your contribution. In accordance with your suggestion, the definitions of CCTA-confirmed CAD and CCTA-confirmed non-CAD have also been added to the Abstract section. These additional parts are shown in bold and italic in the final version of the Abstract provided below. Again, in accordance with your suggestion, the statements regarding the relationships between CACS-detected CAD severity and extent with CTRP5 and chemerin in the CCTA-confirmed CAD group, and that these were conducted as a secondary examination for CAD diagnosis, have been added to both the Abstract and introduction sections (lines 83-88)
Abstract: Background/Objectives: As an endocrine organ, adipose tissue produces adipokines that influence coronary artery disease (CAD). The objective of this study was to assess the potential value of CTRP5 and chemerin in differentiating coronary computed tomography angiography (CCTA)-confirmed coronary artery disease (CAD) versus non-CAD. Secondarily, within the CCTA-confirmed CAD group, the aim was to investigate the relationship between the severity and extent of CAD, as determined by coronary artery calcium score (CACS), and the levels of CTRP5 and chemerin. Methods: Consecutive individuals with chest pain underwent CCTA to evaluate coronary artery anatomy and were divided into two groups. The CCTA-confirmed CAD group included patients with any atherosclerotic plaque (soft, mixed, or calcified) regardless of calcification, while the non-CAD group consisted of individuals without plaques on CCTA, with zero CACS, and without ischemia on stress ECG. Secondarily, in the CCTA-confirmed CAD group, the severity and extent of CAD were evaluated using CACS. Blood samples were collected and stored at -80°C for analysis of CTRP5 and chemerin levels via ELISA. Results: Serum CTRP5 and chemerin levels were significantly higher in the CAD group compared to the non-CAD group (221.83±103.81 vs. 149.35±50.99 ng/mL, p = 0.003 and 105.02±35.62 vs. 86.07±19.47 ng/mL, p = 0.005, respectively). ROC analysis showed that a CTRP5 cutoff of 172.30 ng/mL had 70% sensitivity and 73% specificity for identifying CAD, while a chemerin cutoff of 90.46 ng/mL had 61% sensitivity and 62% specificity. A strong positive correlation was observed between CTRP5 and chemerin, but neither adipokine showed a correlation with the Agatston score, a measure of CAD severity and extent, nor with coronary artery stenosis as determined by CCTA. Conclusions: CTRP5 and chemerin were significantly elevated in the CCTA-confirmed CAD group compared to the non-CAD group, with CTRP5 showing greater sensitivity and specificity. However, neither adipokine was linked to CAD severity and extent, differing from findings based on invasive coronary angiography (ICA). CTRP5 may serve as a promising “all-or-none biomarker” for CAD presence.
Introduction (line 83-88 )
This study sought to assess the possible significance of the adipokine CTRP5, which has been inadequately explored in relation to CAD and the adipokine chemerin in differentiating between individuals with CCTA-confirmed normal coronary arteries, and patients with CCTA-confirmed CAD. The secondary purpose was to examine the correlation between the severity and extent of CAD, as assessed by the coronary artery calcium score (CACS), and serum levels of CTRP5 and chemerin in the CCTA-confirmed CAD group.
Comments 2: My second main comment, and I’m sorry again that I did not make this clear in my initial review, but I understand that there is limited literature about CTRP5 and chemerin and its relationship with CCTA-diagnosed atherosclerosis (hence the need for your study), but reading through your discussion, not all of your key findings seem to be highlighted, and now with the additions of the additional lines in the paragraph, it does not flow well with what a reader would expect to read. The paragraph beginning on line 245 should essentially be your second paragraph, with the discussion regarding CTRP5 following, and then chemerin. Then a discussion about the lack of correlation with the Agaston score is required, and this is when/where you should include the paragraph beginning on line 223. You should consider:
- Lines 213-222: this is not what I expect to read at this point in the discussion. After your opening sentence, I expect to read about a summary of your key findings. What you have written currently belongs in other places around the discussion (particularly before your limitations, as you are essentially summarising your strengths); lines 220-222 may be able to follow-on form the first sentence.
- The paragraph beginning on line 231 should come first, and the line should read ‘In the past, invasive coronary angiography has been employed to evaluate the existence, severity, or stent restenosis of CAD in the majority of studies investigating the correlation between atherosclerosis and CTRP5, chemerin, and other adipokines. This paragraph should also be thought of as a strengths paragraph, and put towards the end of the discussion prior to the limitations
- The paragraph beginning on line 223 should come later in the discussion, and be related to your CACS findings
- The discussion also needs to be read carefully to ensure that there is minimal repetition from beginning to end as it is quite long
Responses 2: Dear Reviewer; as you suggested, the paragraph starting from line 245 has been positioned as the second paragraph. As you suggested, the discussion sections on CTRP5 and then chemerin were included afterwards. Lines 220-222 were written as the second sentence. In lines 213-222, the main results of our study were added to the section where you expressed that you did not expect to read. The paragraph starting at line 231 was shifted towards the end of the discussion as you suggested, and the disadvantages of invasive coronary angiography and the advantages of CCTA were described to illustrate the strengths. Subsequently, as you suggested, the section starting from line 223 was added to explain the reasons for the lack of correlation between CACS, used to define the prevalence and extent of CAD, and CTRP5 and chemerin. Repeating sections in the discussion were removed from the text. In accordance with your suggestion, the reasons for the lack of correlation between CACS and CTRP5 and chemerin were addressed together in the final section, and the repetitive hypotheses were consolidated. The final version of the discussion section is below.
4. Discussion
This study sought to assess differences in plasma levels of the adipokines CTRP5 and chemerin between individuals with CAD, characterized by mixed, calcific, or soft plaques across different stages of CCTA-confirmed atherosclerosis, and those without CAD, confirmed by CCTA. Additionally, the study aimed to investigate the correlation between chemerin and CTRP5 with the CACS (Agatston score), which allows for the objective detection of the severity and extent of CAD in the CCTA-confirmed CAD group [4,25-29]. The findings of our research revealed that the levels of CTRP5 and chemerin were significantly higher in the CCTA-confirmed CAD group compare to the CCTA-confirmed non-CAD group. Our findings indicate that serum CTRP5 levels show better sensitivity and specificity for the diagnosis of CAD compared to serum chemerin levels. A significant positive correlation was observed between serum CTRP5 and chemerin levels. In the CCTA-confirmed CAD group, there was no correlation between the severity and extent of CAD, measured by the CACS (Agatston score), and the serum levels of either adipokine. No statistically significant association was found between serum CTRP5 or chemerin levels and the degree of coronary artery stenosis in the CCTA-confirmed CAD group.
A number of lipids, immune cells, vascular smooth muscle cells (VSMCs), and adipokines are all involved in the complex process that leads to the formation of atherosclerosis. CTRP5, an adipokine that is released from adipose tissue and most commonly from EAT, has the potential to influence the formation of atherosclerotic plaque in a number of different ways [12-17,30]. Li et al. founded that serum CTRP5 concentrations were markedly elevated in patients with CAD relative to individuals with normal coronary arteries, exhibiting a positive correlation with the quantity of affected arteries. The study evaluated atherosclerotic endarterectomy specimens and non-atherosclerotic arteries from CAD patients, demonstrating elevated CTRP5 expression in the endothelium during the early stages of atherosclerosis. Dual immunofluorescence revealed the presence of CTRP5 in endothelial cells, infiltrating macrophages, and VSMCs within the neointima, but absent in the medial layer, exhibiting heightened expression in early atherosclerotic lesions [17]. Another study was performed to examine the biological impacts of CTRP5 on aortic smooth muscle cells (ASMCs). The findings indicated that CTRP5 increased the expression of MMP2, cyclin D1, and TNF-alpha in ASMCs in a dose-dependent manner, while also activating the Notch1, TGF-beta, and hedgehog signaling pathways. The findings indicate that CTRP5 enhances the proliferation, inflammation, and migration of VSMCs via the stimulation of many pathways of signaling in early stages of atherosclerosis [31].Infiltrating macrophages in the artery wall then absorb this altered LDL, resulting in the production of foam cells and heightened inflammatory reactions. 12/15-LOX is crucial for facilitating the absorption and oxidation of LDL in macrophages [30,32-34]. In the previous investigation, Li et al. proposed a theory indicating that 12/15-LOX is modulated by CTRP5 via STAT6 signaling in early stages of atherosclerosis. Additionally, the researchers showed that CTRP5 increased transcytosis of LDL across endothelium mono layers and the oxidative alteration of LDL in the endothelial cells [17]. Liu et al. conducted a study comparing a normal coronary artery group with CAD patients experiencing acute coronary syndrome (ACS). Their findings indicated that, in agreement with our study, CTRP5 levels were significantly elevated in the CAD group compared to the normal coronary artery group, and there was no correlation between the severity and extent of CAD, as assessed by the Gensini Score, and CTRP5 levels [35]. This study significantly contrasts with ours, which utilized the CCTA method, as it utilized invasive coronary angiography (ICA) for diagnosis; established CAD patients with ACS in the CAD cohort; and classified individuals with less than 50% coronary stenosis as part of the normal coronary artery group.
In vitro studies have shown that chemerin stimulation leads to an increase in the production of reactive oxygen species (ROS) and inflammation in human microvascular endothelial cells and VSMCs. This is indicates that chemerin has been shown to exert a substantial influence on vascular dysfunction. The activation of the pro-inflammatory transcriptional regulator NF-κB and the enhancement of monocyte–endothelial adhesion are two of the ways in which it contributes to endothelial inflammation [19,34]. There is a strong connection between chemerin and atherosclerosis because of the impact that it has on macrophages through the chemerin chemokine-like receptor 1 (CMKLR-1) [36]. The expression of chemerin and CMKLR-1 within human aortas, coronary arteries, and periadventitial adipose tissue (PVAT) has been demonstrated to correlate with atherosclerosis, according to immunohistochemical research. Chemerin is found in PVAT, VSMCs, and foam cells within atherosclerotic lesions while CMKLR-1 is expressed in VSMCs and foam cells in atherosclerosis-affected arteries. The levels of expression of chemerin and CMKLR-1 have been found to have significant correlations with the severity of atherosclerosis in the previous research [19,37,38]. Research has demonstrated that chemerin can markedly reduce the production of cyclic guanosine monophosphate (cGMP) and the vasodilatory effects triggered by nitric oxide (NO) [19,39]. Furthermore, chemerin significantly contributes to atherosclerosis formation by facilitating the proliferation and migration of endothelial cells, functioning as a chemoattractant, and stimulating angiogenesis in early stages of atherosclerosis [19,40-42].
This study found that plasma chemerin levels were elevated in patients with CAD compared to those without CAD. This observation is consistent with prior research that have demonstrated a connection between chemerin concentrations and cardiovascular disease [19-22,43,44]. Conversely, our data indicated that there is no link between the severity and extent of CAD and plasma chemerin levels. This finding sharply contrast with prior research that demonstrated a correlation between chemerin levels and the extent and severity of coronary artery disease, as assessed by the Gensini Score derived from ICA results or the number of stenosed coronary arteries. [20,43,44]. The lack of correlation between CAD severity and chemerin plasma levels in our study may be related to the more accurate effectiveness of CCTA in illustrating the comprehensive burden of atherosclerosis and the existence of atherosclerotic plaques at different stages of the disease, relative to the ICA employed for CAD classification in previous investigations [25,26]. There is evidence that chemerin is implicated in the pathogenesis of atherosclerosis, particularly in the early stages of the disease [19,34,36-41].
Invasive coronary angiography was employed to evaluate the existence, severity, or stent restenosis of CAD in the majority of academic studies investigating the correlation between atherosclerosis and CTRP5, chemerin, and other adipokines in the past. The cohort referred to as the CAD group in prior research employing the ICA methodology typically comprised individuals with coronary artery stenosis exceeding 50%. Conversely, patients with stenosis less than 50% were arbitrarily designated as the control group or included in the cohort that had no restenosis following stent placement [17,31,35,43,44]. However, patients exhibiting lower than 50% stenosis or lacking intra luminal stenosis due to atherosclerotic eccentric plaques may not adequately reflect a truly normal cohort free of coronary artery atherosclerosis, particularly in studies employing ICA lumenographic criteria. These earlier studies' control groups contained CAD patients, which adds a level of difficulty to the interpretation of the CAD prediction. The use of ICA to assess the severity of CAD is also problematic because it ignores the existence of eccentric plaques that do not contribute to the load of intraluminal plaque [25,26,31,35,43,44 ].
The present study preferred CCTA, which offers superior accuracy compared to ICA in diagnosing and excluding CAD. Eccentric plaques that do not induce intraluminal narrowing, which are undetectable by ICA, together with minor plaques in the initial stages of atherosclerosis, can be identified by CCTA [4,25,26]. Consequently, in the study, the CCTA-confirmed CAD group, indicative of coronary artery disease, was established with improved precision, while the CCTA-confirmed non-CAD group, representing the true normal coronary artery cohort, was constructed as optimally as possible.
Coronary artery calcification is a defining characteristic of atherosclerosis and a significant contributor to the onset and advancement of CAD. CACS reflects the overall burden of coronary atherosclerosis. Sangiorgi et al. found that coronary calcium measurement effectively assesses atherosclerotic plaque presence and burden, though no strong predictive link was found between luminal narrowing and calcification, possibly due to remodeling [27-29]. The Denmark Heart Registry, using CCTA, showed that total plaque burden “CACS” is a key factor in cardiovascular risk, regardless of stenosis, with similar CACS levels indicating similar event risk, whether CAD is obstructive or not [45].
Contrary to previous studies that identified a correlation between CAD severity and extent, as measured by the Gensini score, SYNTAX score, or the number of coronary arteries with critical stenosis, and serum levels of CTRP5 or chemerin derived from ICA method data [17,31,43,44] our research did not find a relationship between CACS, a marker of CAD severity and extent [27-29], and serum concentrations of CTRP5 and chemerin. Consistent with our findings, Szpakowicz et al. observed no statistically significant link between the SYNTAX Score, which reflects the extent and severity of the disease, and chemerin serum levels in stable CAD patients having percutaneous coronary intervention [46].
The contrasting results may be explained by the advantages of the CCTA method used for defining CAD, which is more effective in detecting early-stage minor and eccentrically located atherosclerotic plaques [25,26] with elevated levels of CTRP5 and chemerin, compared to the ICA method employed in previous studies for CAD diagnosis. The CCTA method employed in this study effectively identifies early-stage plaques in cases of low CAD extent and the absence of intraluminal stenosis. However, CACS, used for assessing disease extent and severity, shows limited effectiveness in detecting early-stage plaques [28,29]. Consequently, patients with coronary artery disease exhibiting varying disease severities and different levels of intraluminal stenosis may present comparable early-stage plaque loads and similar serum levels of CTRP5 and chemerin. The CCTA method allows the identification of patients exhibiting advanced atherosclerosis, as evidenced by increased CACS, significant calcific plaque burden, and lowered early-stage atherosclerotic plaque presence, resulting in reduced levels of CTRP5 and chemerin expression. Conversely, patients with CAD who display elevated levels of CTRP5 and chemerin, minimal CAD extent, low CACS, and significant early-stage atherosclerotic plaque burden (without intraluminal stenosis) may also be identified by CCTA. This may clarify why plasma levels of CTRP5 and chemerin do not correlate with the severity, extent, or CACS of CAD. The absence of a correlation between the severity and extent of CAD, as assessed by CACS, and the levels of CTRP and chemerin in our study may be attributed to the limitations of the CACS method in detecting non-calcified atherosclerotic plaques during the early stages of atherosclerosis, where elevated levels of adipokines are anticipated [17,25,26,31,43,44] .
Minor Comments
Comments 1: Lines 115-118 should be moved to the end of the methods section, but before the statistical analysis section.
Responses 1: Dear Reviewer; As you suggested, lines 115-118 have been placed at the end of the methods section, before the statistical analyses section. In the text below, the section where the sentences from lines 115-118 have been deleted is marked with a strikethrough, and the section where they have been revised is shown in bold italics.
The protocol for the study was approved by the Bakircay University Non-Interventional Clinical Research Ethics Committee (Approved 24th August 2023; Decision number: 1154; Research number: 1155). All individuals who took part in the study provided written informed consent.
Each participant underwent CCTA to determine coronary artery anatomy and the presence or absence of atherosclerotic plaque (CAD) in the coronary arteries. A 128-slice single-source scanner (Somatom Go Top; Siemens Healthcare, Forchheim, Germany) was used for the procedure, and an independent expert was responsible for the evaluation of the results. Using the same CT scanner, the Agatston score was used to determine the degree of CACS.
The protocol for the study was approved by the Bakircay University Non-Interventional Clinical Research Ethics Committee (Approved 24th August 2023; Decision number: 1154; Research number: 1155). All individuals who took part in the study provided written informed consent.
Statistical Analysis:
Comments 2: Lines 119-120 (Each participant… in the coronary arteries.) is not needed as this is implied above.
Responses 2: Dear Reviewer; in accordance with your guideline, the sentence in lines 119-120, which we had repeated, has been removed from the text. The relevant section has been highlighted in the text below.
Each participant underwent CCTA to determine coronary artery anatomy and the presence or absence of atherosclerotic plaque (CAD) in the coronary arteries. A 128-slice single-source scanner (Somatom Go Top; Siemens Healthcare, Forchheim, Germany) was used for the procedure, and an independent expert was responsible for the evaluation of the results. Using the same CT scanner, the Agatston score was used to determine the degree of CACS.
The protocol for the study was approved by the Bakircay University Non-Interventional Clinical Research Ethics Committee (Approved 24th August 2023; Decision number: 1154; Research number: 1155). All individuals who took part in the study provided written informed consent.
Statistical Analysis:
Comments 3: Lines 120-124 should probably go at the end of the paragraph ending on line 102.
Responses 3: Dear reviewer; in accordance with your guidelines, lines 120-124 have been deleted and added to the end of the paragraph ending at line 102.In the text below, the deleted section is shown with a strikethrough, and the section it has been moved to is shown in bold and italics.
In total, there were 106 individuals, and out of those, 89 were determined to be suitable for participation in the study. The CCTA-confirmed CAD group was defined as participants who had any soft, mixed or calcific atherosclerotic plaque in their coronary arteries, whether or not coronary artery calcification (CAC) was present on anatomical examination by CCTA. The CACS, assessed by the Agatston score, was used as an objective measure of the extent and severity of CAD and was used to quantify the extent and severity of CAD in CCTA Confirmed CAD group. Patients with zero CACS but any atherosclerotic plaque detected by CCTA were also included in the CCTA-confirmed CAD group. Participants without atherosclerotic plaque in their coronary arteries, as assessed by CCTA, and exhibiting zero CACS along with no signs of myocardial ischemia on treadmill/stress electrocardiogram or myocardial perfusion scintigraphy—conducted to rule out microvascular disease—were defined as the CCTA-confirmed non-CAD group (true normal coronary artery group). An illustration of the flow chart for the study could be observed in Figure 1.
A 128-slice single-source scanner (Somatom Go Top; Siemens Healthcare, Forchheim, Germany) was used for the procedure, and an independent expert was responsible for the evaluation of the results. Using the same CT scanner, the Agatston score was used to determine the degree of CACS.
Obtaining the patients' consent allowed for the collection of clinical and laboratory data, which included information about the patients' gender, age, height, weight, blood pressure measures, blood cholesterol levels, smoking history, and familial history of CAD. Blood samples were collected from the antecubital vein in the early morning hours of the morning, only after the patient had been fasting for at least eight hours. Following that, a biochemical analyzer was utilized in order to quantify serum lipids in addition to a number of other biochemical parameters. A fasting period of one night was followed by the collection of blood samples for CTRP5 and chemerin, which were then preserved at a temperature of -80 degrees Celsius. Quantification of serum levels of CTRP5 and chemerin was performed with the use of commercially available ELISA kits (Invitrogen ELISA Human C1qTNF5 ELISA Kit and Invitrogen Human RARRES2/TIG2 ELISA Kit) at appropriate dilutions in accordance with the directions provided by the manufacturer.
The protocol for the study was approved by the Bakircay University Non-Interventional Clinical Research Ethics Committee (Approved 24th August 2023; Decision number: 1154; Research number: 1155). All individuals who took part in the study provided written informed consent.
Each participant underwent CCTA to determine coronary artery anatomy and the presence or absence of atherosclerotic plaque (CAD) in the coronary arteries. A 128-slice single-source scanner (Somatom Go Top; Siemens Healthcare, Forchheim, Germany) was used for the procedure, and an independent expert was responsible for the evaluation of the results. Using the same CT scanner, the Agatston score was used to determine the degree of CACS.
The protocol for the study was approved by the Bakircay University Non-Interventional Clinical Research Ethics Committee (Approved 24th August 2023; Decision number: 1154; Research number: 1155). All individuals who took part in the study provided written informed consent.
Statistical Analysis:
Comments 4: Systolic BP in table 1 in the CCTA confirmed CAD group: remove the .0 in the SD value.
Responses 4: Dear Reviewer; we apologize for overlooking this issue, as you pointed out in the first revision. As you suggested earlier, ".0" has been removed from the table. The corrected final version of the table is provided below. the relevant part is written in bold and italic.
AGATSTON |
0 |
224.0 ± 342.3 |
<0.001 |
Systolic Blood Pressure (mmHg) |
135 ± 13 |
132 ± 25 |
0.623 |
Diastolic Blood Pressure (mmHg) |
80 ± 10 |
79 ± 12 |
0.938 |
Comments 5: Table 2. there are commas instead of full stops in the numbers, and all numbers should have leading zeros in them
Responses 5: Dear Reviewer, all decimal numbers in Table 2 have been prefixed with a zero, and commas have been replaced with full stops. The revised version of Table 2 is presented below.
Table 2. Correlations of CTRP5, Chemerin and Agatston score
|
Chemerin006 |
CTRP100 |
AGATSON |
|
Chemerin |
r |
|
0.725** |
0.059 |
P |
|
0.000 |
0.613 |
|
CTRP100 |
r |
0.725** |
1 |
0.061 |
P |
0.000 |
|
0.601 |
|
AGATSON |
r |
0.059 |
0.061 |
1 |
P |
0.613 |
0.601 |
|
Comments 6: Line 217 – CCTA-confirmed CAD group (confirmed needs a lower-case c and similarly non-CAD group on line 218 should be lower-case n (and this should be consistent through the paper)
Responses 6: Dear Reviewer; in accordance with your suggestion, lines 217-218 and the entire text have been reviewed to ensure consistency in the terms "CCTA-confirmed CAD" and "CCTA-confirmed non-CAD." The corrections made in the relevant section are shown in bold and italics.
The findings of our research revealed that the levels of CTRP5 and chemerin were significantly higher in the CCTA-confirmed CAD group compare to the CCTA-confirmed non-CAD group. Our findings indicate that serum CTRP5 levels show better sensitivity and specificity for the diagnosis of CAD compared to serum chemerin levels. A significant positive correlation was observed between serum CTRP5 and chemerin levels. In the CCTA-confirmed CAD group, there was no correlation between the severity and extent of CAD, measured by the CACS (Agatston score), and the serum levels of either adipokine. No statistically significant association was found between serum CTRP5 or chemerin levels and the degree of coronary artery stenosis in the CCTA-confirmed CAD group.

Round 3
Reviewer 2 Report
Comments and Suggestions for Authors
The authors have largely addressed my queries. All the main information is now there, and it is almost ready for publication. However, you now have a very long discussion and needs to be condensed where possible. I would suggest condensing lines 231-255 as there is too much information here, and similarly 265-284, this could also be shortened.
Other minor things that I have picked up in the manuscript include:
- Define the ROC abbreviation in the abstract (line 27)
- Your keys words should largely begin with lower case letters, aside from ‘Agatston’ (this should be upper case)
- Line 132: ‘the’ can be removed from prior to standard deviation
- Is there a reason why Agatston is capitalised in table 1?
- Line 290: the first word should be ‘contrasts’ (not contrast)
Author Response
Major comments:
Comments : The authors have largely addressed my queries. All the main information is now there, and it is almost ready for publication. However, you now have a very long discussion and needs to be condensed where possible. I would suggest condensing lines 231-255 as there is too much information here, and similarly 265-284, this could also be shortened.
Responses1: Dear Reviewer; in accordance with your suggestion, the 306-word section between lines 231-256, which was indeed excessively long, has been summarized while preserving its content, unnecessary information has been removed, and it has been reduced to 171 words. Below, the old version of the text is shown with strikethrough, and the revised final version is written in bold and italics. Thank you very much for your contribution in making the text more readable.
A number of lipids, immune cells, vascular smooth muscle cells (VSMCs), and adipokines are all involved in the complex process that leads to the formation of atherosclerosis. CTRP5, an adipokine that is released from adipose tissue and most commonly from EAT, has the potential to influence the formation of atherosclerotic plaque in a number of different ways [12-17,30]. Li et al. founded that serum CTRP5 concentrations were markedly elevated in patients with CAD relative to individuals with normal coronary arteries, exhibiting a positive correlation with the quantity of affected arteries. The study evaluated atherosclerotic endarterectomy specimens and non-atherosclerotic arteries from CAD patients, demonstrating elevated CTRP5 expression in the endothelium during the early stages of atherosclerosis. Dual immunofluorescence revealed the presence of CTRP5 in endothelial cells, infiltrating macrophages, and VSMCs within the neointima, but absent in the medial layer, exhibiting heightened expression in early atherosclerotic lesions [17]. Another study was performed to examine the biological impacts of CTRP5 on aortic smooth muscle cells (ASMCs). The findings indicated that CTRP5 increased the expression of MMP2, cyclin D1, and TNF-alpha in ASMCs in a dose-dependent manner, while also activating the Notch1, TGF-beta, and hedgehog signaling pathways. The findings indicate that CTRP5 enhances the proliferation, inflammation, and migration of VSMCs via the stimulation of many pathways of signaling in early stages of atherosclerosis [31]. Infiltrating macrophages in the artery wall then absorb this altered LDL, resulting in the production of foam cells and heightened inflammatory reactions. 12/15-LOX is crucial for facilitating the absorption and oxidation of LDL in macrophages [30,32-34]. In the previous investigation, Li et al. proposed a theory indicating that 12/15-LOX is modulated by CTRP5 via STAT6 signaling in early stages of atherosclerosis. Additionally, the researchers showed that CTRP5 increased transcytosis of LDL across endothelium mono layers and the oxidative alteration of LDL in the endothelial cells [17]. The formation of atherosclerosis involves various lipids, immune cells, vascular smooth muscle cells (VSMCs), and adipokines. CTRP5, an adipokine released from adipose tissue, particularly from EAT, may affect the formation of atherosclerotic plaques by multiple mechanisms [12-17, 30]. Li et al. observed that serum CTRP5 levels were significantly higher in patients with CAD compared to individuals with normal coronary arteries, showing a correlation with the number of affected arteries. The study demonstrated elevated CTRP5 expression in the endothelium of early stage atherosclerotic lesions [17]. Another investigation demonstrated that CTRP5 increased the expression of MMP2, cyclin D1, and TNF-alpha in a dose-dependent manner in aortic smooth muscle cells (ASMCs), thereby activating the Notch1, TGF-beta, and hedgehog pathways. This facilitated VSMCs proliferation, inflammation, and migration during the early stages of atherosclerosis [31]. Macrophages within the arterial wall uptake modified LDL, resulting in foam cell formation and increased inflammation. Li et al. proposed that CTRP5 regulates 12/15-LOX through STAT6 signaling, thereby increasing LDL absorption and oxidation in macrophages and endothelial cells [17, 30, 32-34]
Responses 2: Dear Reviewer; the 229-word section located between lines 265-284, which you indicated makes the text difficult to read, has been shortened while preserving its content as you suggested. The section in question has been reduced to 114 words in its final revised form. As you suggested, it has been made easier to read. The modified initial text has been crossed out, and the final version is shown below in bold and italics.
In vitro studies have shown that chemerin stimulation leads to an increase in the production of reactive oxygen species (ROS) and inflammation in human microvascular endothelial cells and VSMCs. This is indicates that chemerin has been shown to exert a substantial influence on vascular dysfunction. The activation of the pro-inflammatory transcriptional regulator NF-κB and the enhancement of monocyte–endothelial adhesion are two of the ways in which it contributes to endothelial inflammation [19,34]. There is a strong connection between chemerin and atherosclerosis because of the impact that it has on macrophages through the chemerin chemokine-like receptor 1 (CMKLR-1) [36]. The expression of chemerin and CMKLR-1 within human aortas, coronary arteries, and periadventitial adipose tissue (PVAT) has been demonstrated to correlate with atherosclerosis, according to immunohistochemical research. Chemerin is found in PVAT, VSMCs, and foam cells within atherosclerotic lesions while CMKLR-1 is expressed in VSMCs and foam cells in atherosclerosis-affected arteries. The levels of expression of chemerin and CMKLR-1 have been found to have significant correlations with the severity of atherosclerosis in the previous research [19,37,38]. Research has demonstrated that chemerin can markedly reduce the production of cyclic guanosine monophosphate (cGMP) and the vasodilatory effects triggered by nitric oxide (NO) [19,39]. Furthermore, chemerin significantly contributes to atherosclerosis formation by facilitating the proliferation and migration of endothelial cells, functioning as a chemoattractant, and stimulating angiogenesis in early stages of atherosclerosis [19,40-42].
In vitro studies indicate that chemerin increases the formation of reactive oxygen species (ROS) and inflammation in human endothelial cells and VSMCs, explaining its contribution to vascular dysfunction. Chemerin promotes the pro-inflammatory NF-κB pathway and enhances endothelial inflammation by promoting monocyte-endothelial adhesion (19,34).Chemerin's role in atherosclerosis related to its interaction with macrophages through the chemerin chemokine-like receptor 1 (CMKLR-1) (36). Immunohistochemical data show a relationship between the expression of chemerin and CMKLR-1 in human arteries and periadventitial adipose tissue, as well as the severity of atherosclerosis (19,37,38). Chemerin contributes to the early stages of atherosclerosis by decreasing cGMP synthesis and nitric oxide-induced vasodilation, enhancing endothelial cell proliferation and migration, and further stimulating angiogenesis (19,39-42).
Minor Comments: Other minor things that I have picked up in the manuscript include:
Comments 1: Define the ROC abbreviation in the abstract (line 27)
Responses 1: Dear Reviewer; The abbreviation ROC in line 27 has been defined. The relevant change is shown below in bold italics.
Receiver operating characteristic (ROC) analysis showed that a CTRP5 cutoff of 172.30 ng/mL had 70% sensitivity and 73% specificity for identifying CAD, while a chemerin cutoff of 90.46 ng/mL had 61% sensitivity and 62% specificity.
Comments 2: Your keys words should largely begin with lower case letters, aside from ‘Agatston’ (this should be upper case)
Responses 2: Dear Reviewer; all the keywords except for the words CTRP5 and Agatston were started with lowercase letters. The relevant section is provided below.
Keywords: CTRP5 ; chemerin ; coronary artery disease ; coronary computed tomography angiography ; Agatston score ; coronary artery calcification ; atherosclerosis ; biomarkers
Comments 3: Line 132: ‘the’ can be removed from prior to standard deviation
Responses 3: Dear Reviewer, in accordance with your suggestion, the "the" before the standard deviation in line 132 has been removed. The relevant part is shown below.
Continuous variables were presented as the mean ± the standard deviation (SD), and the independent samples t-test was used to examine the data.
Continuous variables were presented as the mean ± standard deviation (SD), and the independent samples t-test was used to examine the data
Comments 4: Is there a reason why Agatston is capitalised in table 1?
Responses 4: Dear Reviewer; the spelling of "AGATSTON" in Table 1 has been corrected to "Agatston." The relevant section of the table is shown below.
BMI (kg/m2 ) |
28.1 ± 3.4 |
27.6 ± 3.1 |
0.535 |
Metabolic Syndrome % |
25.7 |
27.7 |
0.756 |
Agatston score |
0 |
224.0 ± 342.3 |
<0.001 |
Systolic Blood Pressure (mmHg) |
135 ± 13 |
132 ± 25 |
0.623 |
Diastolic Blood Pressure (mmHg) |
80 ± 10 |
79 ± 12 |
0.938 |
Comments 5: Line 290: the first word should be ‘contrasts’ (not contrast)
Responses 5: Dear reviewer; The word "contrast" in line 290 has been changed to "contrasts" in accordance with your suggestion. The relevant section is shown below in bold and italics.
Conversely, our data indicated that there is no link between the severity and extent of CAD and plasma chemerin levels. This finding sharply contrasts with prior research that demonstrated a correlation between chemerin levels and the extent and severity of coronary artery disease, as assessed by the Gensini Score derived from ICA results or the number of stenosed coronary arteries. [20,43,44].
